# Non-stationary Diffusion For Probabilistic Time Series Forecasting

Weiwei Ye [1][*]   Zhuopeng Xu [1][*]   Ning Gui [1]

## Abstract

Due to the dynamics of underlying physics and external influences, the uncertainty of time series varies over time. However, existing Denoising Diffusion Probabilistic Models (DDPMs) fail to capture this non-stationary nature, constrained by their constant variance assumption from the additive noise model (ANM). In this paper, we innovatively utilize the Location-Scale Noise Model (LSNM) to relax the fixed uncertainty assumption of ANM. A diffusion-based probabilistic forecasting framework, termed Non-stationary Diffusion (NsDiff), is designed based on LSNM that is capable of modeling the changing pattern of uncertainty. Specifically, NsDiff combines a denoising diffusion-based conditional generative model with a conditional mean and a variance estimator, enabling adaptive endpoint distribution modeling. Furthermore, we propose an uncertainty-aware noise schedule, which dynamically adjusts the noise levels to accurately reflect the data uncertainty at each step and integrates the time-varying variances into the diffusion process. Extensive experiments conducted on nine real-world and synthetic datasets demonstrate the superior performance of NsDiff compared to existing approaches. Code is available at `https://github.com/wwy155/NsDiff`.

## 1. Introduction

Time series forecasting plays a key role in various fields such as traffic prediction (Ermagun & Levinson, 2018) and supply chain management (Chopra & Meindl, 2021). Given a historical multivariate series $\mathbf{X}$, general forecasting methods involve training an $f(\mathbf{X})$ to predict a future series $\mathbf{Y}$, which can be viewed as modeling the $\mathbb{E}[\mathbf{Y}|\mathbf{X}]$. Although recent research has demonstrated promising capabilities to model conditional expectations (Zhou et al., 2021; Wu et al., 2020), effective decision-making, particularly in high-stakes fields like healthcare (Bertozzi et al., 2020) and finance (Li & Bastos, 2020), often requires accurately estimating the uncertainty underlying the data (Kendall & Gal, 2017; Xu et al., 2024). To address this problem, many recent studies have focused on probabilistic time series forecasting (Rasul et al., 2021; Chen et al., 2024; Li et al., 2024b), where the goal to estimate a distribution of possible future outcomes along with their associated uncertainties.

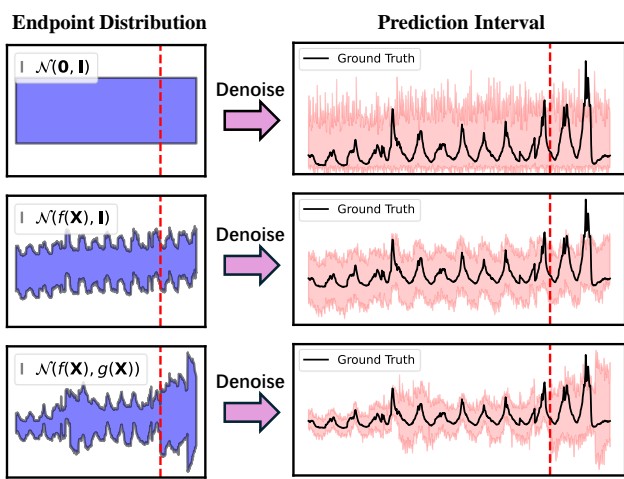

*Figure 1.* A figure illustrates DDPMs with different endpoints trained to estimate the number of influenza-like disease patients weekly. We plot the endpoint distributions and prediction intervals of $\mathcal{N}(0, \mathbf{I})$ (Top), $\mathcal{N}(f(\mathbf{X}), \mathbf{I})$ (Middle), and $\mathcal{N}(f(\mathbf{X}), g(\mathbf{X}))$ (Bottom) on the left and right, respectively. The red dashed line indicates the division of the training and test dataset.

The Denoising Diffusion Probabilistic Models (DDPMs) have recently gained significant attention for probabilistic time series forecasting due to their powerful ability to generate high-dimensional data (Rasul et al., 2021; Tashiro et al., 2021; Li et al., 2024a). Existing DDPMs typically rely on the Additive Noise Model (ANM) (Spirtes et al., 2001), which assumes $\mathbf{Y} = f(\mathbf{X}) + \boldsymbol{\epsilon}$, where $\boldsymbol{\epsilon} \sim \mathcal{N}(\mathbf{0}, \boldsymbol{\sigma})$ represents stationary Gaussian noise. The primary objective of these models is not only to estimate the conditional expectation $\mathbb{E}[\mathbf{Y}|\mathbf{X}]$ via $f(\mathbf{X})$, but also to accurately capture uncertainty by modeling the noise distribution $\boldsymbol{\epsilon}$. While DDPMs with stationary Gaussian noise have achieved sub-

---

[*]Equal contribution [1]School of Computer Science and Engineering, Central South University, Changsha, China. Correspondence to: Ning Gui <ninggui@gmail.com>.

*Proceedings of the 42nd International Conference on Machine Learning*, Vancouver, Canada. PMLR 267, 2025. Copyright 2025 by the author(s).

stantial success in domains such as computer vision and natural language generation (Ho et al., 2020; Dhariwal & Nichol, 2021; Gu et al., 2022), they are less effective for modeling non-stationary time series data, where patterns of uncertainty vary contextually (Lee et al., 2024).

Figure 1 illustrates an example from the ILI (influenza-like illness) dataset, with different endpoint distributions (Left) and estimated uncertainty (Right) on different models: TimeGrad (Rasul et al., 2021), TMDM (Li et al., 2024b), and NsDiff (ours). In the upper part of Figure 1, TimeGrad (Rasul et al., 2021) employs the endpoint $\mathcal{N}(\mathbf{0}, \mathbf{I})$, which fails to capture non-stationary characteristics. TMDM (Li et al., 2024a) uses $\mathcal{N}(f(\mathbf{X}), \mathbf{I})$ as endpoint, representing changing averages. On the test dataset (shown to the right of the red dashed line), where both the number of patients and the corresponding deviation increase, the performance differences are evident. TimeGrad fails to model both the underlying trends and deviations. In contrast, TMDM effectively captures the trends through its $f(\mathbf{X})$, but its stationary covariance $\mathbf{I}$ limits its ability to accurately estimate uncertainty, which is critical for the probabilistic time series forecasting.

To better address non-stationarity with changing uncertainty, we introduce Location-Scale Noise Model (LSNM) into DDPMs, which relaxes the traditional Additive Noise Model (ANM) by incorporating a contextually changing variance: $\mathbf{Y} = f(\mathbf{X}) + \sqrt{g(\mathbf{X})}\epsilon$, where $g(\mathbf{X})$ is an $\mathbf{X}$-dependent variance model and $\epsilon$ is a standard gaussian noise. LSNM is capable of modeling both the contextual mean through $f(\mathbf{X})$ and the contextual uncertainty through $\sqrt{g(\mathbf{X})}$. In the special case where $g(\mathbf{X}) \equiv 1$, this simplifies to the standard ANM. Building upon this more flexible and expressive assumption, we propose the **N**on-**s**tationary **Diff**usion Model (**NsDiff**) framework, which provides an uncertainty-aware noise schedule for diffusion process. In summary, our contributions are:

- We observe that the ANM is inadequate for capturing the varying uncertainty and propose a novel framework that integrates LSNM to allow for explict uncertainty modeling. This work is the first attempt to introduce LSNM into probabilistic time series forecasting.

- To fundamentally elevate the noise modeling capabilities of DDPM, we seamlessly integrate time-varying variances into the core diffusion process through an uncertainty-aware noise schedule that dynamically adapts the noise variance at each step.

- Experimental results indicate that **NsDiff** achieves superior performance in capturing uncertainty. Specifically, in comparison to the second-best recent baseline TMDM, NsDiff improves up to 66.3% on real-world datasets and 88.3% on synthetic datasets.

## 2. Related Works

### 2.1. DDPM for Probabilistic Forecasting

Denoising Diffusion Probabilistic Models (DDPMs) have shown promising results in the probabilistic forecasting area (Tyralis & Papacharalampous, 2022). Rasul et al. (2021) introduce TimeGrad, an autoregressive diffusion model guided by a recurrent neural network hidden state. Tashiro et al. (2021) propose a masking strategy for training diffusion models, applicable to tasks like imputation and forecasting. Alcaraz & Strodthoff (2022) extend DDPMs with a structured space model to capture long-term dependencies. TimeDiff (Shen & Kwok, 2023) utilized future mixup and autoregressive initialization. Li et al. (2022b) integrate multiscale denoising score matching to guide the diffusion process, ensuring generated series align with the target. DiffusionTS (Yuan & Qiao, 2024) trains the model to reconstruct the sample rather than noise, using a Fourier-based loss term. Kollovieh et al. (2024) propose a self-guiding strategy for time series generation and forecasting based on structured state-space models. By leveraging the bridge-based model introduced by Shi et al. (2023), Chen et al. (2023) present a convergence analysis of the Schrödinger bridge algorithm and propose improvements to the diffusion process. However, these methods generally assume fixed endpoint variance, which is hard to model non-stationary time series.

### 2.2. Non-stationary Time Series Forecasting

To address non-stationarity, Li et al. (2022a) employ a domain-adaptation approach to predict data distributions, while Du et al. (2021) propose an adaptive RNN for distribution matching to mitigate non-stationary effects. Liu et al. (2022) introduce a non-stationary Transformer with de-stationary attention to account for non-stationary factors in self-attention. Wang et al. (2022) use global and local Koopman operators to capture patterns at different scales, and Liu et al. (2024a) apply Koopman operators to components identified via Fourier transforms. Other approaches decompose stationary and non-stationary parts, such as Ogasawara et al. (2010) with local normalization, and Passalis et al. (2019) with a learnable, instance-wise normalization. RevIN (Kim et al., 2021) addresses the distribution shift using reversible normalization, and recent works (Fan et al., 2023; Liu et al., 2024b) explore finer-grained trend modeling. Jiang et al. (2023) addresses non-stationarity in chaotic systems by preserving invariant measures to stabilize dynamical systems over time without relying on domain-specific priors. Fourier transforms, closely linked with non-stationarity, are also been applied to tackle these issues (Fan et al., 2024; Ye et al., 2024). Despite these advances in time series forecasting, the non-stationary uncertainty in probabilistic forecasting remains largely unexplored.

## 3. Preliminary

### 3.1. Problem Formulation

Given a historical multivariate time series $\mathbf{X} \in \mathbb{R}^{N \times D}$ where $N$ is the historical window size and $D$ denotes the number of feature dimensions. The probabilistic forecasting task is to predict the distribution of the future multivariate time series $\mathbf{Y} = \{p(\boldsymbol{y}_1), p(\boldsymbol{y}_2), ..., p(\boldsymbol{y}_M)|\boldsymbol{y} \in \mathbb{R}^D\}$, where $M$ is the future window size. While previous works model the future series with ANM: $\mathbf{Y} = f_\phi(\mathbf{X}) + \boldsymbol{\epsilon}$, we model it based on LSNM with a more generalized data model:

$$\mathbf{Y} = f_\phi(\mathbf{X}) + \sqrt{g_\psi(\mathbf{X})}\boldsymbol{\epsilon} \tag{1}$$

where the $\boldsymbol{\epsilon} \sim \mathcal{N}(0, \boldsymbol{\sigma})$ is Gaussian noise. The $f_\phi(\mathbf{X})$ and $g_\psi(\mathbf{X})$ can be viewed as prior knowledge with pre-trained parameters $\phi$ and $\psi$, where the $f_\phi(\mathbf{X})$ is modeling the conditional expectation $\mathbb{E}[\mathbf{Y}|\mathbf{X}]$ and $g_\psi(\mathbf{X})$ is modeling the varying uncertainty. In this paper, we incorporate this two prior knowledge into the diffusion model to tackle the non-stationary challenge in probabilistic time series forecasting.

### 3.2. Denoising Diffusion Probabilistic Models

DDPMs (Ho et al., 2020) is a popular generative model to estimate the uncertainty for future time series. In the original DDPM, the future series distribution can be represented as $p_\theta(\mathbf{Y}_0) := \int p_\theta(\mathbf{Y}_{0:T})d\mathbf{Y}_{1:T}$, where $\mathbf{Y}_1, ..., \mathbf{Y}_T$ are latent variables. The joint distribution is defined as a Markov chain $p_\theta(\mathbf{Y}_{0:T}) := p(\mathbf{Y}_T) \prod_{t=1}^T p_\theta(\mathbf{Y}_{t-1}|\mathbf{Y}_t)$, where the endpoint of diffusion is set to $p(\mathbf{Y}_T) := \mathcal{N}(0, \mathbf{I})$. To generate the distribution $p_\theta(\mathbf{Y}_0)$, DDPM designs two processes: a forward process to gradually add noise and a reverse process to denoise. In the forward process, the future series $\mathbf{Y}_0$ gradually diffuses to the given prior endpoint $\mathbf{Y}_T$ without any trainable parameters.

$$q(\mathbf{Y}_{1:T}|\mathbf{Y}_0) := \prod_{t=1}^T q(\mathbf{Y}_t|\mathbf{Y}_{t-1})$$
$$q(\mathbf{Y}_t|\mathbf{Y}_{t-1}) := \mathcal{N}(\mathbf{Y}_t; \sqrt{1-\beta_t}\mathbf{Y}_{t-1}, \beta_t\mathbf{I}) \tag{2}$$

where $\beta_t \in (0, 1)$ is a diffusion schedule for controlling the endpoint $\mathbf{Y}_T \sim \mathcal{N}(0, \mathbf{I})$. This forward sampling can be simplified by $q(\mathbf{Y}_t|\mathbf{Y}_0) = \mathcal{N}(\mathbf{Y}_t; \sqrt{\bar{\alpha}_t}\mathbf{Y}_0, (1-\bar{\alpha}_t)\mathbf{I})$ in practice, where $\alpha_t := 1 - \beta_t$ and $\bar{\alpha}_t := \prod_{i=1}^t \alpha_i$.

The reverse process parameterizes $p_\theta(\mathbf{Y}_{t-1}|\mathbf{Y}_t)$ and compares it against forward process posteriors $q(\mathbf{Y}_{t-1}|\mathbf{Y}_t, \mathbf{Y}_0)$. DDPM has shown that matching these two posteriors is equivalent to estimating the added noise $\boldsymbol{\eta}$ in the forward process. Thus, the parameterization of $p_\theta(\mathbf{Y}_{t-1}|\mathbf{Y}_t)$ is:

$$p_\theta(\mathbf{Y}_{t-1}|\mathbf{Y}_t) := \mathcal{N}(\mathbf{Y}_{t-1}; \boldsymbol{\mu}_\theta(\mathbf{Y}_t, t), \frac{1-\bar{\alpha}_{t-1}}{1-\bar{\alpha}_t}\beta_t\mathbf{I})$$
$$\boldsymbol{\mu}_\theta(\mathbf{Y}_t, t) := \frac{1}{\sqrt{\alpha_t}}(\mathbf{Y}_t - \frac{\beta_t}{\sqrt{1-\bar{\alpha}_t}}\boldsymbol{\eta}_\theta)) \tag{3}$$

where $\boldsymbol{\eta}_\theta$ is the estimated noise by a denoising model, which optimizes the following objective:

$$\mathbb{E}_{\mathbf{Y}_0 \sim q(\mathbf{Y}_0), \boldsymbol{\eta} \sim \mathcal{N}(0, \mathbf{I}), t}||\boldsymbol{\eta} - \boldsymbol{\eta}_\theta||^2 \tag{4}$$

Following this basic forward and reverse process, many diffusion-based methods improve the reverse process (Rasul et al., 2021; Shen & Kwok, 2023) or prior distribution (Li et al., 2024a) with the historical time series information. However, they fix the variance of the prior distribution and focus on the expectation matching. The prior setup and training of uncertainty are largely ignored.

## 4. Methodology

In this section, we introduce the proposed NsDiff, including the design of forward and reverse process distributions, as well as the training and inference procedures of NsDiff. Furthermore, we discuss two simplified versions of NsDiff. The outline of NsDiff is given in Figure 2.

### 4.1. Forward and Reverse Process

In previous diffusion-based methods, the uncertainty prior was missing, and they tended to set the endpoint of diffusion $\mathbf{Y}_T$ to $\mathcal{N}(0, \mathbf{I})$ or $\mathcal{N}(f_\phi(\mathbf{X}), \mathbf{I})$. To improve this, we use a different noise model LSNM to form the endpoint:

$$p(\mathbf{Y}_T|f_\phi(x), g_\psi(x)) := \mathcal{N}(f_\phi(\mathbf{X}), g_\psi(\mathbf{X})) \tag{5}$$

where $f_\phi(\mathbf{X})$ models the conditional expectation $\mathbb{E}[\mathbf{Y}|\mathbf{X}]$ which can be parameterized by any forecasting model, e.g., Dlinear (Zeng et al., 2023) or PatchTST (Nie et al., 2022). We follow previous works (Kim et al., 2021; Liu et al., 2024b) to train the prior scale of uncertainty $g_\psi(\mathbf{X})$. We use the input variance to predict the output variance.

The forward process incrementally modifies the noise at each step to approach the endpoint distribution. To seamlessly integrate time-varying variances into the diffusion process, we propose an uncertainty-aware noise schedule, and incorporate data variance into the forward process distribution $q(\mathbf{Y}_t \mid \mathbf{Y}_{t-1}, f_\phi(\mathbf{X}), g_\psi(\mathbf{X}), \boldsymbol{\sigma}_{\mathbf{Y}_0})$. Specifically, given well-pretrained models $f_\phi, g_\psi$, and a prior state $\mathbf{Y}_{t-1}$, we control the scaled variance to transition from the actual variance $\boldsymbol{\sigma}_{\mathbf{Y}_0}$ at the starting point to the endpoint $g_\psi(\mathbf{X})$. The resulting distribution is normally distributed as:

$$\mathcal{N}(\mathbf{Y}_t; \sqrt{\alpha_t}\mathbf{Y}_{t-1} + (1-\sqrt{\alpha_t})f_\phi(\mathbf{X}),$$
$$\underbrace{(\beta_t^2 g_\psi(\mathbf{X}) + \alpha_t\beta_t\boldsymbol{\sigma}_{\mathbf{Y}_0}))}_{\boldsymbol{\sigma}_t} \tag{6}$$

where the shared coefficient $\beta_t$ is a noise scaling constant. As the noise step $t$ increases, the term $\beta_t g_\psi(\mathbf{X})$ grows and $\alpha_t\boldsymbol{\sigma}_{\mathbf{Y}_0}$ decreases. At $t = T$, where $\alpha_t = 0$, the variance converges to the assumed endpoint $g_\phi(\mathbf{X})$.

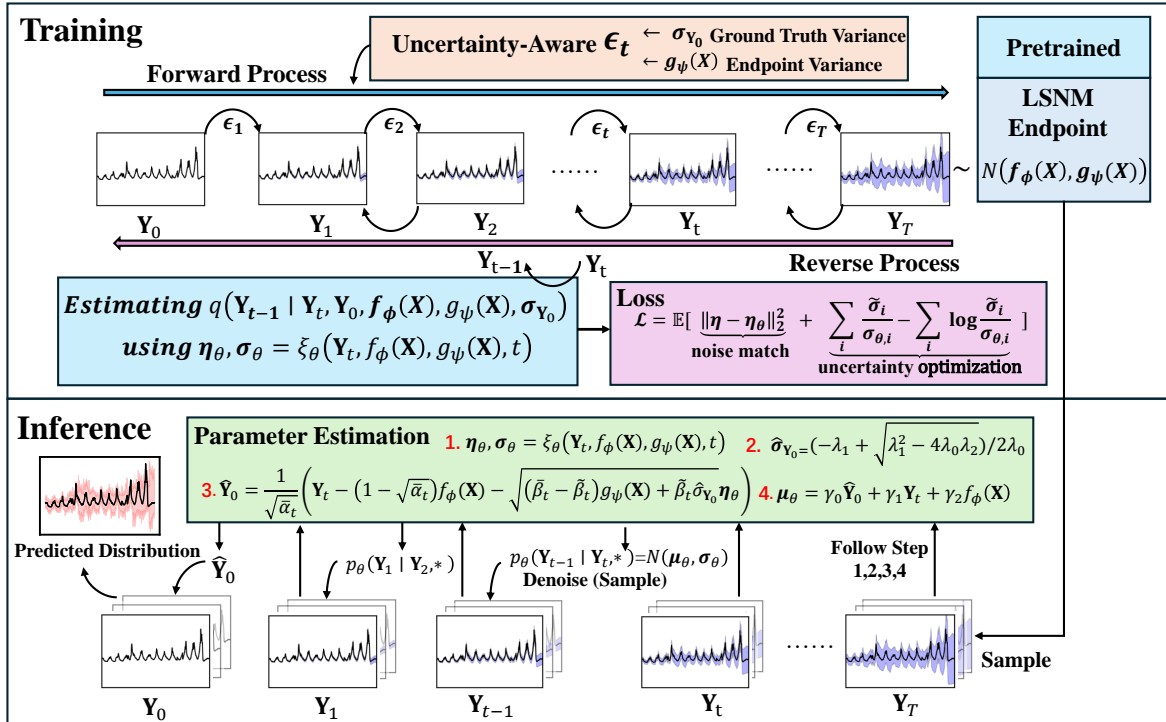

*Figure 2.* The outline of NsDiff. It integrates a LSNM-based endpoint and an uncertainty-aware noise schedule. During the training phase, a noise and variance estimator, $\xi_\theta$, is optimized to approximate the reverse process distribution. During inference, it samples from LSNM endpoint and use the estimated reverse distribution to iteratively denoise and generate the final prediction.

This enables DDPM to adaptively adjust the noise levels at each step to capture the data uncertainty. The forward distribution admits a closed-form sampling distribution $q(\mathbf{Y}_t|\mathbf{Y}_0, f_\phi(\mathbf{X}), g_\psi(\mathbf{X}), \boldsymbol{\sigma}_{\mathbf{Y}_0})$ with an arbitrary timestep $t$:

$$\mathcal{N}(\mathbf{Y}_t; \sqrt{\bar{\alpha}_t}\mathbf{Y}_0 + (1 - \sqrt{\bar{\alpha}_t})f_\phi(\mathbf{X}),$$
$$\underbrace{(\bar{\beta}_t - \tilde{\beta}_t)g_\psi(\mathbf{X}) + \tilde{\beta}_t\boldsymbol{\sigma}_{\mathbf{Y}_0}}_{\bar{\boldsymbol{\sigma}}_t}) \quad (7)$$

where we define the following coefficients:

$$\tilde{\alpha}_t := \sum_{k=0}^{t-1}\prod_{i=t-k}^{t}\alpha_i, \quad \bar{\beta}_t := 1 - \bar{\alpha}_t$$
$$\hat{\alpha}_t := \sum_{k=0}^{t-1}\left(\prod_{i=t-k}^{t}\alpha_i\right)\alpha_{t-k}, \quad \tilde{\beta}_t := \tilde{\alpha}_t - \hat{\alpha}_t. \quad (8)$$

we leave the detailed derivation of $\bar{\boldsymbol{\sigma}}_t$ to Appendix A.1, and all these coefficients are positive numbers. Notably, under a perfect estimator (assuming $g_\psi(\mathbf{X}) = \boldsymbol{\sigma}_{\mathbf{Y}_0}$), $\bar{\boldsymbol{\sigma}}_t$ simplifies to $\bar{\beta}_t g_\phi(\mathbf{X})$, and with the additional assumption of $\boldsymbol{\sigma}_{\mathbf{Y}_0} = \mathbf{I}$, it degenerates to the earlier constant variance settings ($\bar{\beta}_t\mathbf{I}$). More detailed discussions and derivations can be found in Section 4.6 and Appendix A.5.

In the reverse process, the posteriors of $\mathbf{Y}_{t-1}$ are tractable when conditioned on $\mathbf{Y}_0$, which can be restated as:

$$q(\mathbf{Y}_{t-1}|\mathbf{Y}_t, \mathbf{Y}_0, f_\phi(\mathbf{X}), g_\psi(\mathbf{X}), \boldsymbol{\sigma}_{\mathbf{Y}_0}) := \mathcal{N}(\mathbf{Y}_{t-1}; \tilde{\boldsymbol{\mu}}, \tilde{\boldsymbol{\sigma}}) \quad (9)$$

where

$$\tilde{\boldsymbol{\mu}} := \gamma_0\mathbf{Y}_0 + \gamma_1\mathbf{Y}_t + \gamma_2 f_\phi(\mathbf{X}) \quad (10)$$

$$\tilde{\boldsymbol{\sigma}} := \frac{\boldsymbol{\sigma}_t\bar{\boldsymbol{\sigma}}_{t-1}}{\alpha_t\bar{\boldsymbol{\sigma}}_{t-1} + \boldsymbol{\sigma}_t} \quad (11)$$

and $\gamma_{0,1,2}$ in $\tilde{\boldsymbol{\mu}}$ are given as:

$$\gamma_0 := \frac{\sqrt{\bar{\alpha}_{t-1}}\boldsymbol{\sigma}_t}{\alpha_t\bar{\boldsymbol{\sigma}}_{t-1} + \boldsymbol{\sigma}_t}, \quad \gamma_1 := \frac{\sqrt{\alpha_t}\bar{\boldsymbol{\sigma}}_{t-1}}{\alpha_t\bar{\boldsymbol{\sigma}}_{t-1} + \boldsymbol{\sigma}_t}$$
$$\gamma_2 := \frac{\sqrt{\alpha_t}(\alpha_t - 1)\bar{\boldsymbol{\sigma}}_{t-1} + (1 - \sqrt{\bar{\alpha}_{t-1}})\boldsymbol{\sigma}_t}{\alpha_t\bar{\boldsymbol{\sigma}}_{t-1} + \boldsymbol{\sigma}_t} \quad (12)$$

We leave the derivation in Appendix A.2. We follow the basic step of DDPM to parameterize a denoise model $p_\theta(\mathbf{Y}_{t-1}|\mathbf{Y}_t, f_\phi(\mathbf{X}), g_\psi(\mathbf{X}))$ to match the forward process posteriors $q(\mathbf{Y}_{t-1}|\mathbf{Y}_t, f_\phi(\mathbf{X}), g_\psi(\mathbf{X}), \boldsymbol{\sigma}_{\mathbf{Y}_0})$.

## 4.2. Loss Function

We approximate the denoising transition step $p_\theta(\mathbf{Y}_{t-1}|\mathbf{Y}_t, f_\phi(\mathbf{X}), g_\psi(\mathbf{X}))$ to the ground-truth de-

noising transition step $q(\mathbf{Y}_{t-1}|\mathbf{Y}_t, f_\phi(\mathbf{X}), g_\psi(\mathbf{X}), \boldsymbol{\sigma}_{\mathbf{Y}_0})$ by optimizing the KL divergence (Hershey & Olsen, 2007) between the posterior distribution $q$ and the parametrized distribution $p_\theta$. Like classic DDPM, we optimize only the diagonal variance term, denoted as $\tilde{\boldsymbol{\sigma}}$ and $\boldsymbol{\sigma}_\theta$ respectively. The loss is defined as the KL divergence of the noise matching term:

$$\mathcal{L} = \mathbb{E}\left[D_{\mathrm{KL}}\left(\mathcal{N}\boldsymbol{x}; \tilde{\boldsymbol{\mu}}, \tilde{\boldsymbol{\sigma}}\|\mathcal{N}\left(\boldsymbol{y}; \boldsymbol{\mu_\theta}, \boldsymbol{\sigma_\theta}\right)\right)\right]$$
$$\propto \mathbb{E}\left[||\boldsymbol{\eta} - \boldsymbol{\eta}_\theta||_2^2 + \sum_i \frac{\tilde{\boldsymbol{\sigma}}_i}{\boldsymbol{\sigma}_{\theta,i}} - \sum_i \log\left(\frac{\tilde{\boldsymbol{\sigma}}_i}{\boldsymbol{\sigma}_{\theta,i}}\right)\right]$$
$$(13)$$

where $\boldsymbol{\eta}_\theta$ is the estimated noise and $\boldsymbol{\eta}$ is the ground truth noise. The first term ensures the estimation of the posterior mean, while the rest terms guarantee the estimation of the variance. We provide the proof in Appendix A.3.

### 4.3. Pretraining $f_\phi$ and $g_\psi$

To train $f_\psi$, we follow prior work (Li et al., 2024a) and utilize the Non-stationary Transformer (Liu et al., 2022) as the backbone model. The training process is identical to that of standard supervised time series models (Zhou et al., 2021). For the training of $g_\psi(\mathbf{X})$, we use a sliding window approach to extract the estimated ground truth variance, similar to references (Kim et al., 2021; Liu et al., 2024b; Ye et al., 2024). Specifically, given time series label $\mathbf{Y}_0$ the estimated ground truth variance is defined as:

$$\boldsymbol{\sigma}_{\mathbf{Y}_0} = \mathrm{Var}(\mathrm{SlidingWindow}(\mathbf{Y}_0)) \qquad (14)$$

thus, the training of $g_\psi(\mathbf{X})$ is formulated as a supervised task. In our implementation, we utilize a sliding stride of 1 and a window size of 96. The function $g_\psi$ is implemented as a three-layer MLP, with outputs passed through the softplus activation (Zheng et al., 2015) to ensure positivity. Further implementation details can be found in Appendix C.2 and we examine the necessity of pretraining in Appendix B.2.

### 4.4. Training NsDiff

The target of NsDiff training is to match posterior distribution $q$ by parameterizing $p_\theta$. Like traditional DDPM, NsDiff can be trained end-to-end by sampling a random $t$ and noise $\boldsymbol{\eta}$ from uniform and Gaussian distributions respectively. According to Eq. 13, we build an estimation model $\xi_\theta(\mathbf{Y}_t, f_\phi(\mathbf{X}), g_\psi(\mathbf{X}), t)$ during the training process to match the noise and variance. The overall procedure is presented in Algorithm 1.

### 4.5. Inference

The target of the inference phase is to recursively sample from the parameterized distribution $p_\theta(\mathbf{Y}_{t-1} \mid \mathbf{Y}_t, f_\phi(\mathbf{X}), g_\psi(\mathbf{X}))$, we provide a detailed

---

**Algorithm 1** Training

**Input:** Data $\mathbf{X}$, target $\mathbf{Y}$, model $f_\phi$, noise and variance estimation model $\xi_\theta$, total timesteps $T$
Pre-train $f_\phi(\mathbf{X})$ to predict $\mathbb{E}(\mathbf{Y}|\mathbf{X})$
Pre-train $g_\psi(\mathbf{X})$ to predict $\mathrm{Var}(\mathbf{Y}|\mathbf{X})$
**repeat**
   Draw $\mathbf{Y}_0 \sim q(\mathbf{Y}_0 \mid \mathbf{X})$
   Draw $t \sim \mathrm{Uniform}(\{1, \ldots, T\})$
   Draw $\boldsymbol{\eta} \sim \mathcal{N}(\mathbf{0}, \mathbf{I})$
   Compute $\mathbf{Y}_t$:
   $\mathbf{Y}_t = \sqrt{\bar{\alpha}_t}\mathbf{Y}_0 + (1 - \sqrt{\bar{\alpha}_t})f_\phi(\mathbf{X})$
      $+ \sqrt{(\bar{\beta}_t - \tilde{\beta}_t)g_\psi(\mathbf{X}) + \tilde{\beta}_t\boldsymbol{\sigma}_{\mathbf{Y}_0}}\boldsymbol{\eta}$ $\triangleright$ using Eq. 7
   Compute estimated noise and variance:
   $\boldsymbol{\eta}_\theta, \boldsymbol{\sigma}_\theta = \xi_\theta(\mathbf{Y}_t, f_\phi(\mathbf{X}), g_\psi(\mathbf{X}), t)$
   Compute loss $\mathcal{L}$        $\triangleright$ using Eq. 13
   Numerical optimization step on $\nabla_\theta\mathcal{L}$
**until** Convergence

---

process in Algorithm 2. At the inference phase, according to Eq. 10 and 11, the calculation of the parameters for the reverse distribution requires estimating both $\mathbf{Y}_0$ and $\boldsymbol{\sigma}_{\mathbf{Y}_0}$. For the estimation of $\mathbf{Y}_0$, we follow prior work (Han et al., 2022) and utilize the relationship between $\mathbf{Y}_t$ and $\mathbf{Y}_0$ as defined in Eq. 7. However, the estimation of $\boldsymbol{\sigma}_{\mathbf{Y}_0}$ lacks a direct correspondence with $\mathbf{Y}_t$. To estimate $\boldsymbol{\sigma}_{\mathbf{Y}_0}$, one straightforward approach is to directly use $g_\psi(\mathbf{X})$. However, it demands a perfect predictor and does not incorporate the reverse process into parameter estimation. Actually, Eq. 11 can be expanded as a quadratic equation with respect to $\boldsymbol{\sigma}_{\mathbf{Y}_0}$. Thus, we utilize the quadratic expansion of Eq. 11 to approximate $\boldsymbol{\sigma}_{\mathbf{Y}_0}$, we leave the detailed derivation at Appendix A.4. Specifically, expanding Eq. 11 gives the following solvable equation:

$$\lambda_0\boldsymbol{\sigma}_{\mathbf{Y}_0}^2 + \lambda_1\boldsymbol{\sigma}_{\mathbf{Y}_0} + \lambda_2 = 0 \qquad (15)$$

where the coefficients are

$$\lambda_0 := \alpha_t\beta_t\tilde{\beta}_{-1}$$
$$\lambda_1 := \beta_t^2\tilde{\beta}_{-1} + \alpha_t\beta_t(\bar{\beta}_{-1} - \tilde{\beta}_{-1})g_\psi(\mathbf{X}) -$$
$$\boldsymbol{\sigma}_\theta(\alpha_t\tilde{\beta}_{-1} + \alpha_t\beta_t)) \qquad (16)$$
$$\lambda_2 := g_\psi(\mathbf{X})^2\beta_t^2(\bar{\beta}_{-1} - \tilde{\beta}_{-1}) -$$
$$\boldsymbol{\sigma}_\theta g_\psi(\mathbf{X})(\alpha_t\bar{\beta}_{-1} - \alpha_t\tilde{\beta}_{-1} + \beta_t^2)$$

$\lambda_0$ is a positive value, and according to Vieta's theorem (Lang, 2012), when $\lambda_2 < 0$, the equation has exactly one positive root. The constraint for $\lambda_2 < 0$ is equivalent to:

$$g_\psi(\mathbf{X}) < \boldsymbol{\sigma}_\theta\left(\frac{\alpha_t}{\beta_t^2} + \frac{1}{\bar{\beta}_{-1} - \tilde{\beta}_{-1}}\right) \qquad (17)$$

Therefore, the solvability of the equation is governed by the noise level parameter $\beta_t$. Under the typical DDPM parameterization (Ho et al., 2020), where $\beta_t$ ranges from 0.0001

**Algorithm 2** Inference

**Input:** data $\mathbf{X}$, models $f_\phi$, $g_\psi$, and $\xi_\theta$
Initialize $\mathbf{Y}_T \sim \mathcal{N}(f_\phi(\mathbf{X}), g_\psi(\mathbf{X}))$
**for** $t = T$ **to** $1$ **do**
  **if** $t > 1$ **then**
    Draw $\mathbf{z} \sim \mathcal{N}(\mathbf{0}, \mathbf{I})$
  **end if**
  Compute $\boldsymbol{\eta}_\theta, \boldsymbol{\sigma}_\theta = \xi_\theta(\mathbf{Y}_t, f_\phi(\mathbf{X}), g_\psi(\mathbf{X}), t)$
  Compute $\hat{\boldsymbol{\sigma}}_{\mathbf{Y}_0} = \frac{-\lambda_1 + \sqrt{\lambda_1^2 - 4\lambda_0\lambda_2}}{2\lambda_0}$   ▷ using Eq. 18
  Compute $\hat{\mathbf{Y}}_0 = \frac{1}{\sqrt{\bar{\alpha}_t}}\left( \mathbf{Y}_t - (1 - \sqrt{\bar{\alpha}_t}) f_\phi(\mathbf{X}) - \right.$
  $\left. \sqrt{(\bar{\beta}_t - \tilde{\beta}_t)g_\psi(\mathbf{X}) + \tilde{\beta}_t\hat{\boldsymbol{\sigma}}_{\mathbf{Y}_0}}\,\boldsymbol{\eta}_\theta \right)$   ▷ using Eq. 7
  **if** $t > 1$ **then**
    Set $\mathbf{Y}_{t-1} = \gamma_0\hat{\mathbf{Y}}_0 + \gamma_1\mathbf{Y}_t + \gamma_2 f_\phi(\mathbf{X}) + \sqrt{\sigma_\theta}\mathbf{z}$
  **else**
    Set $\mathbf{Y}_{t-1} = \hat{\mathbf{Y}}_0$
  **end if**
**end for**
**Output:** $\mathbf{Y}_0$

to 0.02, the coefficient on the right-hand side of Eq. 17 becomes sufficiently large, thereby ensuring the equation's solvability. Hence, by solving the quadratic equation in Eq. 15, we can estimate the value of $\sigma_{\mathbf{Y}_0}$ during inference stage, the specific formula is given by Eq. 18:

$$\hat{\boldsymbol{\sigma}}_{\mathbf{Y}_0} = \frac{-\lambda_1 + \sqrt{\lambda_1^2 - 4\lambda_0\lambda_2}}{2\lambda_0} \tag{18}$$

Experimentally, the approach exhibits consistent solvability across all datasets. We provide more discussions in Appendix A.4.

### 4.6. Simplified Variants of NsDiff

In this section, we discuss two simplified versions of NsDiff by simplifying the variance terms in Eq. 7. We summarize these two variants in Table 1, and provide the ablation results in Section 5.3.

*Table 1.* NsDiff Variants.

| Variants | Endpoint | Forward Noise |
|---|---|---|
| w/o LSNM | $\mathcal{N}f_\phi(\mathbf{X}), \mathbf{I})$ | $\beta_t\mathbf{I}$ |
| w/o UANS | $\mathcal{N}(f_\phi(\mathbf{X}), g_\psi(\mathbf{X}))$ | $\beta_t g_\psi(\mathbf{X})$ |
| NsDiff | $\mathcal{N}(f_\phi(\mathbf{X}), g_\psi(\mathbf{X}))$ | $\beta_t^2 g_\psi(\mathbf{X}) + \beta_t\alpha_t\boldsymbol{\sigma}_{\mathbf{Y}_0}$ |

**Perfect Estimator (w/o UANS)**: Assuming a perfect variance estimator $g_\psi(\mathbf{X}) = \boldsymbol{\sigma}_{\mathbf{Y}_0}$, Eq. 6 becomes the following:

$$\mathcal{N}(\mathbf{Y}_t; \sqrt{\alpha_t}\mathbf{Y}_{t-1} + (1 - \sqrt{\alpha_t})f_\phi(\mathbf{X}), \\ (1 - \alpha_t)\,g_\psi(\mathbf{X})) \tag{19}$$

Further derivations show that this is simply a constant multiplication of the variance term from prior works (Han et al., 2022), and the training of the variance is not necessary. However, this estimation of uncertainty has two main drawbacks. First, assuming a perfect estimator inherently introduces bias. In addition, this approach estimates the variance without leveraging the denoising process, as the variance is fully determined by pretrained $g_\psi(\mathbf{X})$.

**Unit Variance (w/o LSNM)**: Assuming a known unit variance, i.e., $g_\psi(\mathbf{X}) = \boldsymbol{\sigma}_{\mathbf{Y}_0} = \mathbf{I}$, Eq. 6 becomes:

$$\mathcal{N}(\mathbf{Y}_t; \sqrt{\alpha_t}\mathbf{Y}_{t-1} + (1 - \sqrt{\alpha_t})f_\phi(\mathbf{X}), (1 - \alpha_t)\,\mathbf{I}). \tag{20}$$

which is consistent with previous work (Han et al., 2022). TMDM (Li et al., 2024a) is a typical probabilistic forecasting model built under this assumption. The results for TMDM are presented in Section 5.2 and 5.3, where we conduct experiments on real and synthetic datasets, respectively. We provide detailed derivations and more discussions in Appendix A.5.

## 5. Experiments

### 5.1. Experiment Setup

**Datasets**: Nine popular real-world datasets with diverse characteristics are selected, including Electricity (ECL), ILI, ETT{h1, h2, m1, m2}, ExchangeRage (EXG), Traffic, and SolarEnergy (Solar). Table 2 summarizes basic statistics for these datasets. To estimate uncertainty variation between the train and test datasets, we use the ratio of test variance to train variance, selecting the highest value across dimensions to capture non-stationary uncertainty. A detailed notebook on this calculation is available in our repository. For dataset splits, we follow previous time series prediction works (Wu et al., 2022; Li et al., 2024b): the ETT datasets are split 12/4/4 months for train/val/test, while others are split 7:1:2. Details can be found in Appendix C.1.

*Table 2.* Dataset properties, including total dimentions, total timsteps, prediction steps, evaluated uncertainty variation.

| Dataset | Dim. | Steps | Pred.steps | Uncert.Var. |
|---|---|---|---|---|
| ETTm1 | 7 | 69,680 | 192 | 2.53 |
| ETTm2 | 7 | 69,680 | 192 | 1.27 |
| ETTh1 | 7 | 17420 | 192 | 2.50 |
| ETTh2 | 7 | 17,420 | 192 | 1.29 |
| EXG | 8 | 7,588 | 192 | 0.85 |
| ILI | 7 | 966 | 36 | 8.26 |
| ECL | 321 | 26,304 | 192 | 3.94 |
| Traffic | 862 | 17,544 | 192 | 181.83 |
| Solar | 137 | 52,560 | 192 | 0.92 |

**Baselines**: We selected five strong probabilistic forecasting baselines for comparison, including TimeGrad (Rasul et al., 2021), CSDI (Tashiro et al., 2021), TimeDiff (Shen

*Table 3.* Experiment result on nine real-world datasets, **bold face** indicate best result.

| Models | Datasets | ETTh1 | ETTh2 | ETTm1 | ETTm2 | ECL | EXG | ILI | Solar | Traffic |
|---|---|---|---|---|---|---|---|---|---|---|
| TimeGrad | CRPS | 0.606 | 1.212 | 0.647 | 0.775 | 0.397 | 0.826 | 1.140 | **0.293** | 0.407 |
| (2021) | QICE | 6.731 | 9.488 | 6.693 | 6.962 | 7.118 | 9.464 | 6.519 | 7.378 | 4.581 |
| CSDI | CRPS | 0.492 | 0.647 | 0.524 | 0.817 | 0.577 | 0.855 | 1.244 | 0.432 | 1.418 |
| (2022) | QICE | 3.107 | 5.331 | 2.828 | 8.106 | 7.506 | 7.864 | 7.693 | 9.957 | 13.613 |
| TimeDiff | CRPS | 0.465 | 0.471 | 0.464 | 0.316 | 0.750 | 0.433 | 1.153 | 0.700 | 0.771 |
| (2023) | QICE | 14.931 | 14.813 | 14.795 | 13.385 | 15.466 | 14.556 | 14.942 | 14.914 | 15.439 |
| DiffusionTS | CRPS | 0.603 | 1.168 | 0.574 | 1.035 | 0.633 | 1.251 | 1.612 | 0.470 | 0.668 |
| (2024) | QICE | 6.423 | 9.577 | 5.605 | 9.959 | 8.205 | 10.411 | 10.090 | 6.627 | 5.958 |
| TMDM | CRPS | 0.452 | 0.383 | 0.375 | 0.289 | 0.461 | 0.336 | 0.967 | 0.350 | 0.557 |
| (2024) | QICE | 2.821 | 4.471 | 2.567 | 2.610 | 10.562 | 6.393 | 6.217 | 9.342 | 10.676 |
| NsDiff | CRPS | **0.392** | **0.358** | **0.346** | **0.256** | **0.290** | **0.324** | **0.806** | 0.300 | **0.378** |
| **(ours)** | QICE | **1.470** | **2.074** | **2.041** | **2.030** | **6.685** | **5.930** | **5.598** | **6.820** | **3.601** |

& Kwok, 2023), TMDM (Li et al., 2024a) and Diffu-sionTS (Yuan & Qiao, 2024). Specifically, TMDM denoises from $\mathcal{N}(f_\phi(\mathbf{X}), \mathbf{I})$ while others denoise from $\mathcal{N}(\mathbf{0}, \mathbf{I})$.

**Experiment Settings**: Experiments are conducted under popular long-term multivariate forecasting settings, using an input length of 168 in all experiments. All experiments are run with seeds $\{1, 2, 3\}$ for 10 epochs. We use the best result from the validation set to evaluate the model on the test set. The learning rate is set to 0.001, batch size of 32 and the number of timesteps $T = 20$, consistent with prior work (Rasul et al., 2021). We employ a linear noise schedule with $\beta^1 = 10^{-4}$ and $\beta^T = 0.02$, in line with the setup used in conventional DDPM (Ho et al., 2020). At inference, we generate 100 samples to estimate the distribution. For the baseline models, we utilize their default parameters.

**Metrics**: Following prior work (Li et al., 2024a), we use two probabilistic forecasting metrics: Quantile Interval Coverage Error (QICE) (Han et al., 2022) and Continuous Ranked Probability Score (CRPS) (Matheson & Winkler, 1976). For both metrics, smaller values indicate better performance. Detailed formula is provided in Appendix C.3. We provide point forecast results at Appendix B.1.

### 5.2. Main Experiments

To evaluate the performance of NsDiff in probabilistic multivariate time series forecasting, we tested it on nine real-world datasets and compared it to five competitive baselines. The results, summarized in Table 3, show that NsDiff consistently achieves state-of-the-art (SOTA) performance, with superior uncertainty estimation capabilities, except on the Solar dataset, which exhibits low uncertainty variation (0.92 shown in Table 2). Compared to the second-best and previous SOTA TMDM, which uses an endpoint distribution of $\mathcal{N}(f_\phi(\mathbf{X}), \mathbf{I})$, NsDiff demonstrates significant improvements, particularly in the uncertainty interval estimation

metric (QICE). For example, QICE is reduced by 47.9% on ETTh1, 53.6% on ETTh2, 20.5% on ETTm1, and 66.3% on Traffic. Notably, on the Traffic dataset, which has the highest uncertainty variation (181.83), NsDiff achieves the most significant improvement, underscoring its strength in handling high-uncertainty scenarios.

**Sample Showcases** To provide a clearer understanding of NsDiff's performance, we visualize a sample from the ETTh1 dataset in Figure 3. As shown, NsDiff effectively captures the uncertainty, even under the distribution shift between input and output. In contrast, TMDM, while capable of detecting mean variations, fails to adequately model the uncertainty due to its assumption of uncertainty invariance. Other models, such as TimeGrad, CSDI, and TimeDiff, which begin denoising from $\mathcal{N}(\mathbf{0}, \mathbf{I})$, struggle to capture both the mean and variance. For example, as seen on the right side of the figure, TimeGrad predicts a stable trend instead of the observed downward shift. This highlights the limitations of these models in handling non-stationary behavior. In contrast, NsDiff excels at modeling such non-stationary dynamics while providing accurate uncertainty estimation, demonstrating its robustness and effectiveness in challenging forecasting scenarios. We provide other showcases in Appendix D.

### 5.3. Experiments On Synthetic Data

To accurately evaluate NsDiff's performance under time-varying conditions, we designed two synthetic datasets using the LSNM. Specifically, the formula used is $\mathbf{Y} = \mathbf{m}[\mathbf{t}] + \mathbf{v}[\mathbf{t}]\epsilon$, where $m$ and $v$ defines the level of trend and uncertainty variation. In the linear setting, $\mathbf{m}$ increases linearly from 1 to 10, and $\mathbf{v}$ follows the same pattern. In contrast, for the quadratic setting, $\mathbf{v}$ grows quadratically from 1 to 100. The total length of the generated dataset is 7588, and we predict univariate features. The results of these experiments are summarized in Table 4. Further details about

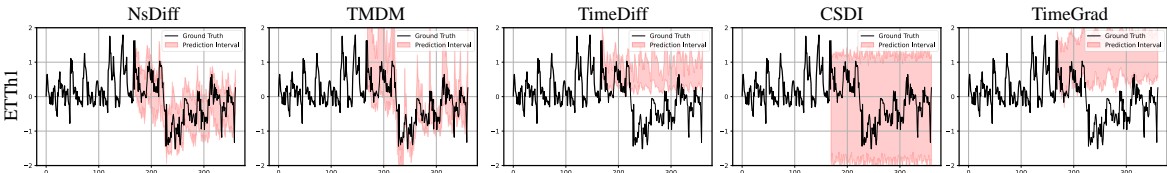

*Figure 3.* The 95% prediction intervals of a ETTh1 sample, the black line is the true values, the red area represents the prediction interval.

the dataset construction can be found in Appendix C.1.2.

*Table 4.* Performace comparison for synthetic datasets

| Variance | Linear | | Quadratic | |
| --- | --- | --- | --- | --- |
| Models | CRPS | QICE | CRPS | QICE |
| TimeGrad | 1.129 | 3.669 | 2.204 | 10.740 |
| CSDI | 1.100 | 3.332 | 1.866 | 5.050 |
| TimeDiff | 1.274 | 10.314 | 2.495 | 14.670 |
| DiffusionTS | 1.454 | 9.290 | 2.123 | 11.273 |
| TMDM | 1.111 | 4.542 | 2.217 | 11.404 |
| NsDiff | **1.057** | **0.987** | **1.777** | **1.336** |

As shown in Table 4, NsDiff achieves remarkable performance under conditions with varying variance. Compared to the previous model, TMDM, in terms of QICE, NsDiff improves performance by 78.3% on the linear-growing variance dataset, and this improvement increases to 88.3% on the quadratic-growing variance dataset. These results demonstrate the superior performance of NsDiff in capturing uncertainty shifts.

**Synthetic Dataset Showcases.** To visually illustrate whether NsDiff can capture the uncertainty shift between the training and test datasets, we provide an example of a linear synthetic dataset in Figure 4, where the estimations for training and extended testing samples are plotted. As shown in the figure, both TMDM and NsDiff effectively capture the uncertainty within the training set. However, in the testing area (to the right of the red dashed line), TMDM assumes invariant uncertainty, while NsDiff successfully captures the uncertainty shift. This clearly demonstrates that NsDiff effectively captures the distribution shift between the training and test datasets, whereas previous methods under ANM fail to do so.

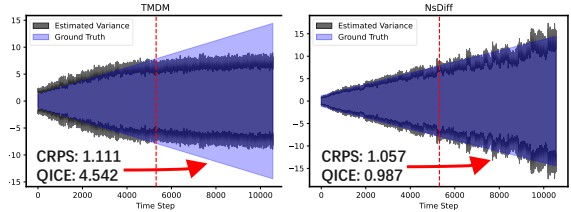

*Figure 4.* The estimated variance and ground truth in linear variance dataset, the variance is estimated using 100 samples. The red dashed line indicates the split of training and extended test sets.

### 5.4. Ablation Experiments

This section compares two simplified variants of NsDiff discussed in Section 4.6, the ablation experiments are conducted on ETTh1 dataset. The abaltion variants are : (1) w/o LSNM: without LSNM assumption, which assumes conditional unit constant variance ($\sigma_{\mathbf{Y}_0} = \mathbf{I}$) (2) w/o UANS: without uncertainty-aware noise schedule, which assumes a perfect noise estimator ($\sigma_{\mathbf{Y}_0} = g_\psi(\mathbf{X})$). The results, presented in Table 5, show that NsDiff achieves the best performance, not only in overall metrics but also in the stability of results (lower variance). Notably, while assuming a perfect uncertainty estimator (w/o UANS) improves CRPS by introducing variable uncertainty, it remains suboptimal in QICE compared to w/o LSNM and exhibits higher variance. This is likely due to potential overfitting of the variance estimator, as it fully relies on $g_\psi(\mathbf{X})$. These findings highlight the importance of a controllable noise schedule, rather than solely relying on a perfect $g_\psi(\mathbf{X})$.

*Table 5.* Variants information and ablation experiment results.

| Metrics | Forward Noise | QICE | CRPS |
| --- | --- | --- | --- |
| w/o LSNM | $\beta_t \mathbf{I}$ | 2.821±0.718 | 0.452±0.027 |
| w/o UANS | $\beta_t g_\psi(\mathbf{X})$ | 3.184±0.787 | 0.413±0.015 |
| NsDiff | $\beta_t^2 g_\psi(\mathbf{X}) + \beta_t \alpha_t \sigma_{\mathbf{Y}_0}$ | **1.470±0.207** | **0.392±0.009** |

## 6. Conclusion

In this paper, we present Non-stationary Diffusion (NsDiff), a novel class of conditional Denoising Diffusion Probabilistic Models (DDPMs) specifically designed to advance probabilistic forecasting. NsDiff represents the first attempt to integrate the Location-Scale Noise Model (LSNM) into probabilistic forecasting, providing a more flexible and expressive framework for uncertainty representation in the data. We introduce an uncertainty-aware noise schedule, which enhances the noise modeling capabilities of DDPMs by incorporating time-varying variances directly into the diffusion process. NsDiff provides a generalized framework that extends the flexibility of existing models; by incorporating a pretrained mean and variance estimator along with the designed noise schedule, NsDiff enables accurate uncertainty estimation, thereby opening new opportunities for advancing research in probabilistic forecasting.

## Acknowledgements

This work is supported in part by the National Natural Science Foundation of China (Grant No. 6247075381) and partially funded by Huaneng Headquarters Technology Projects with No. HNKJ23-HF97. We would also like to thank the anonymous reviewers for their constructive feedback and suggestions.

## Impact Statement

This paper presents work whose goal is to advance the field of Machine Learning. There are many potential societal consequences of our work, none which we feel must be specifically highlighted here.

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

# A. Deriviation

## A.1. Closed-form for Forward Process Distribution of $\mathbf{Y}_0$

Following the original DDPM, we first $\alpha_t := 1 - \beta_t$ and $\bar{\alpha}_t := \prod_{i=1}^{t} \alpha_i$, where $\beta_t \in (0,1)$ is a diffusion schedule. To simplify the deriviation, we further define $\bar{\beta}_t := 1 - \bar{\alpha}_t$ and $\boldsymbol{\sigma}_t = (1 - \alpha_t)^2 g_\psi(\mathbf{X}) + (1 - \alpha_t)\alpha_t \sigma_{\mathbf{Y}_0}$. With the forward distribution in Eq. 6, expanding the forward process starting from $t$ to $0$ gives the following deriviation:

$$
\begin{aligned}
\mathbf{Y}_t &= \sqrt{\alpha_t}\mathbf{Y}_{t-1} + (1 - \sqrt{\alpha_t}) f_\phi(\mathbf{X}) + \sqrt{\boldsymbol{\sigma}_t}\boldsymbol{\eta}_t, \\
&= \sqrt{\alpha_t}\left[\sqrt{\alpha_{t-1}}\mathbf{Y}_{t-2} + (1 - \sqrt{\alpha_{t-1}}) f_\phi(\mathbf{X}) + \sqrt{\boldsymbol{\sigma}_{t-1}}\boldsymbol{\eta}_{t-1}\right] + (1 - \sqrt{\alpha_t}) f_\phi(\mathbf{X}) + \sqrt{\boldsymbol{\sigma}_t}\boldsymbol{\eta}_t, \\
&= \sqrt{\alpha_t\alpha_{t-1}}\mathbf{Y}_{t-2} + \sqrt{\alpha_t}(1 - \sqrt{\alpha_{t-1}}) f_\phi(\mathbf{X}) + (1 - \sqrt{\alpha_t}) f_\phi(\mathbf{X}) + \sqrt{\alpha_t}\sqrt{\boldsymbol{\sigma}_{t-1}}\boldsymbol{\eta}_{t-1} + \sqrt{\boldsymbol{\sigma}_t}\boldsymbol{\eta}_t, \\
&= \sqrt{\alpha_t\alpha_{t-1}}\mathbf{Y}_{t-2} + (1 - \sqrt{\alpha_t\alpha_{t-1}}) f_\phi(\mathbf{X}) + \sqrt{\alpha_t\boldsymbol{\sigma}_{t-1} + \boldsymbol{\sigma}_t}\boldsymbol{\eta}_{t-1}, \\
&= \sqrt{\alpha_t\alpha_{t-1}}\left[\sqrt{\alpha_{t-2}}\mathbf{Y}_{t-3} + (1 - \sqrt{\alpha_{t-2}})f_\phi(\mathbf{X}) + \sqrt{\boldsymbol{\sigma}_{t-2}}\boldsymbol{\eta}_{t-2})\right] + (1 - \sqrt{\alpha_t\alpha_{t-1}}) f_\phi(\mathbf{X}) + \sqrt{\alpha_t\boldsymbol{\sigma}_{t-1} + \boldsymbol{\sigma}_t}\boldsymbol{\eta}_{t-1}, \\
&= \sqrt{\alpha_t\alpha_{t-1}\alpha_{t-2}}\mathbf{Y}_{t-3} + (1 - \sqrt{\alpha_t\alpha_{t-1}\alpha_{t-2}}) f_\phi(\mathbf{X}) + \sqrt{\alpha_t\alpha_{t-1}\boldsymbol{\sigma}_{t-2} + \alpha_t\boldsymbol{\sigma}_{t-1} + \boldsymbol{\sigma}_t}\boldsymbol{\eta}_{t-2}, \\
&\cdots
\end{aligned}
$$

$$
\mathbf{Y}_t = \sqrt{\alpha_t\alpha_{t-1}\cdots\alpha_1}\mathbf{Y}_0 + [1 - \sqrt{\alpha_t\alpha_{t-1}\cdots\alpha_1}] f_\phi(\mathbf{X}) + \sqrt{\sum_{k=0}^{t-1}\left(\prod_{j=t-k+1}^{t}\alpha_j\right)\boldsymbol{\sigma}_{t-k}}\boldsymbol{\eta}_0
\tag{21}
$$

Eq. 21 describes the relationship between $\mathbf{Y}_t$ and $\mathbf{Y}_0$. To simplify the notation, we further give the following definition:

$$
\boldsymbol{\sigma}_t = (\alpha_t^2 - \alpha_t + (1 - \alpha_t))g_\psi(\mathbf{X}) + (\alpha_t - \alpha_t^2)\boldsymbol{\sigma}_{\mathbf{Y}_0}
\tag{22}
$$

$$
\sum_{k=0}^{t-1}\left(\prod_{j=t-k+1}^{t}\alpha_j\right)(1 - \alpha_{t-k}) = (1 - \alpha_t) + \alpha_t(1 - \alpha_{t-1}) + (\alpha_t\alpha_{t-1})(1 - \alpha_{t-2}) + \ldots = 1 - \prod_{i=1}^{t}\alpha_i
\tag{23}
$$

$$
\sum_{k=0}^{t-1}\left(\prod_{j=t-k+1}^{t}\alpha_j\right)\alpha_{t-k} = \alpha_t + \alpha_t\alpha_{t-1} + \alpha_t\alpha_{t-1}\alpha_{t-2} + \ldots = \sum_{k=0}^{t-1}\prod_{i=t-k}^{t}\alpha_i
\tag{24}
$$

$$
\sum_{k=0}^{t-1}\left(\prod_{j=t-k+1}^{t}\alpha_j\right)\alpha_{t-k}^2 = \alpha_t^2 + \alpha_t\alpha_{t-1}^2 + \alpha_t\alpha_{t-1}\alpha_{t-2}^2 + \ldots = \sum_{k=0}^{t-1}(\prod_{i=t-k}^{t}\alpha_i)\alpha_{t-k}
\tag{25}
$$

where we define above Eq. 23, 24 and 25 as $\bar{\alpha}_t$, $\tilde{\alpha}_t$, $\hat{\alpha}_t$ respectively. Eq. 21 becomes the following form:

$$
\mathbf{Y}_t = \sqrt{\bar{\alpha}_t}\mathbf{Y}_0 + (1 - \sqrt{\bar{\alpha}_t})f_\phi(\mathbf{X}) + \underbrace{\sqrt{(\hat{\alpha}_t - \tilde{\alpha}_t + 1 - \bar{\alpha}_t)g_\psi(\mathbf{X}) + (\tilde{\alpha}_t - \hat{\alpha}_t)\boldsymbol{\sigma}_{\mathbf{Y}_0}}}_{\sqrt{\bar{\boldsymbol{\sigma}}_t}}\boldsymbol{\eta}_0
\tag{26}
$$

$$
= \sqrt{\bar{\alpha}_t}\mathbf{Y}_0 + (1 - \sqrt{\bar{\alpha}_t})f_\phi(\mathbf{X}) + \underbrace{\sqrt{(\bar{\beta}_t - \tilde{\beta}_t)g_\psi(\mathbf{X}) + \tilde{\beta}_t\boldsymbol{\sigma}_{\mathbf{Y}_0}}}_{\sqrt{\bar{\boldsymbol{\sigma}}_t}}\boldsymbol{\eta}_0
\tag{27}
$$

where we define $\tilde{\beta}_t = \tilde{\alpha}_t - \hat{\alpha}_t$ and $\bar{\boldsymbol{\sigma}}_t = (\bar{\beta}_t - \tilde{\beta}_t)g_\psi(\mathbf{X}) + \tilde{\beta}_t\boldsymbol{\sigma}_{\mathbf{Y}_0}$. Eq. 27 gives the relationship between $\mathbf{Y}_t$ and $\mathbf{Y}_0$, which admits the closed-form sampling distribution given $\mathbf{Y}_0$ with an arbitrary timestep $t$.

## A.2. Reverse Posterior Distribution

To simplify the notation of the following derivation, we first give the following definition:

$$
\mathbf{A} = \mathbf{Y}_t - (1 - \sqrt{\alpha_t}) f_\phi(\mathbf{X})
\tag{28}
$$

$$
\mathbf{B} = \sqrt{\bar{\alpha}_{t-1}}\mathbf{Y}_0 + (1 - \sqrt{\bar{\alpha}_{t-1}}) f_\phi(\mathbf{X})
\tag{29}
$$

With above definition, the conditional distribution of reverse process is:

$$q\left(\mathbf{Y}_{t-1} \mid \mathbf{Y}_t, \mathbf{Y}_0, \mathbf{X}\right) \propto q\left(\mathbf{Y}_t \mid \mathbf{Y}_{t-1}, f_\phi(\mathbf{X}), g_\psi(\mathbf{X})\right) q\left(\mathbf{Y}_{t-1} \mid \mathbf{Y}_0, f_\phi(\mathbf{X}), g_\psi(\mathbf{X})\right)$$

$$\propto \exp\left(-\frac{1}{2}\left(\frac{\left(\mathbf{A}-\sqrt{\alpha_t}\mathbf{Y}_{t-1}\right)^2}{\boldsymbol{\sigma}_t}+\frac{(\mathbf{Y}_{t-1}-\mathbf{B})^2}{\bar{\boldsymbol{\sigma}}_{t-1}}\right)\right)$$

$$= \exp\left(-\frac{1}{2}\left(\frac{\mathbf{A}^2-2\sqrt{\alpha_t}\mathbf{A}\mathbf{Y}_{t-1}+\alpha_t(\mathbf{Y}_{t-1})^2}{\boldsymbol{\sigma}_t}+\frac{(\mathbf{Y}_{t-1})^2-2\mathbf{B}\mathbf{Y}_{t-1}+\mathbf{B}^2}{\bar{\boldsymbol{\sigma}}_{t-1}}\right)\right) \qquad (30)$$

$$\propto \exp\left(-\frac{1}{2}\left(\frac{\alpha_t}{\boldsymbol{\sigma}_t}(\mathbf{Y}_{t-1})^2-\frac{2\sqrt{\alpha_t}\mathbf{A}}{\boldsymbol{\sigma}_t}\mathbf{Y}_{t-1}+\frac{1}{\bar{\boldsymbol{\sigma}}_{t-1}}(\mathbf{Y}_{t-1})^2-\frac{2\mathbf{B}}{\bar{\boldsymbol{\sigma}}_{t-1}}\mathbf{Y}_{t-1}\right)\right)$$

$$= \exp\left(-\frac{1}{2}\left(\frac{\alpha_t}{\boldsymbol{\sigma}_t}+\frac{1}{\bar{\boldsymbol{\sigma}}_{t-1}}\right)(\mathbf{Y}_{t-1})^2-2\left(\frac{\sqrt{\alpha_t}\mathbf{A}}{\boldsymbol{\sigma}_t}+\frac{\mathbf{B}}{\bar{\boldsymbol{\sigma}}_{t-1}}\right)\mathbf{Y}_{t-1}\right)$$

then, the parameter $\tilde{\mu}$ in the posteriors of $\mathbf{Y}_{t-1}$ is equivalent to:

$$\tilde{\boldsymbol{\mu}} = \frac{\frac{\sqrt{\alpha_t}\mathbf{A}}{\boldsymbol{\sigma}_t}+\frac{\mathbf{B}}{\bar{\boldsymbol{\sigma}}_{t-1}}}{\frac{\alpha_t}{\boldsymbol{\sigma}_t}+\frac{1}{\bar{\boldsymbol{\sigma}}_{t-1}}} = \frac{\sqrt{\alpha_t}\mathbf{A}\bar{\boldsymbol{\sigma}}_{t-1}+\mathbf{B}\boldsymbol{\sigma}_t}{\alpha_t\bar{\boldsymbol{\sigma}}_{t-1}+\boldsymbol{\sigma}_t}$$

$$= \frac{\sqrt{\alpha_t}\left(\mathbf{Y}_t-\left(1-\sqrt{\alpha_t}\right)f_\phi(\mathbf{X})\right)\bar{\boldsymbol{\sigma}}_{t-1}+\left(\sqrt{\bar{\alpha}_{t-1}}\mathbf{Y}_0+\left(1-\sqrt{\bar{\alpha}_{t-1}}\right)f_\phi(\mathbf{X})\right)\boldsymbol{\sigma}_t}{\alpha_t\bar{\boldsymbol{\sigma}}_{t-1}+\boldsymbol{\sigma}_t} \qquad (31)$$

$$= \left(\underbrace{\frac{\sqrt{\bar{\alpha}_{t-1}}\boldsymbol{\sigma}_t}{\alpha_t\bar{\boldsymbol{\sigma}}_{t-1}+\boldsymbol{\sigma}_t}}_{\gamma_0}\right)\mathbf{Y}_0 + \left(\underbrace{\frac{\sqrt{\alpha_t}\bar{\boldsymbol{\sigma}}_{t-1}}{\alpha_t\bar{\boldsymbol{\sigma}}_{t-1}+\boldsymbol{\sigma}_t}}_{\gamma_1}\right)\mathbf{Y}_t + \left(\underbrace{\frac{\sqrt{\alpha_t}(\alpha_t-1)\bar{\boldsymbol{\sigma}}_{t-1}+(1-\sqrt{\bar{\alpha}_{t-1}})\boldsymbol{\sigma}_t}{\alpha_t\bar{\boldsymbol{\sigma}}_{t-1}+\boldsymbol{\sigma}_t}}_{\gamma_2}\right)f_\phi(\mathbf{X})$$

where, the specific form can be written as:

$$\gamma_0 = \frac{\sqrt{\bar{\alpha}_{t-1}}\boldsymbol{\sigma}_t}{\alpha_t\bar{\boldsymbol{\sigma}}_{t-1}+\boldsymbol{\sigma}_t} = \frac{\sqrt{\bar{\alpha}_{t-1}}(\beta_t^2 g_\psi(\mathbf{X})+\alpha_t\beta_t\boldsymbol{\sigma}_{\mathbf{Y}_0})}{g_\psi(\mathbf{X})(\alpha_t\bar{\beta}_{t-1}-\alpha_t\tilde{\beta}_{t-1}+\beta_t^2)+\boldsymbol{\sigma}_{\mathbf{Y}_0}(\alpha_t\tilde{\beta}_{t-1}+\alpha_t\beta_t)}$$

$$\gamma_1 = \frac{\sqrt{\alpha_t}\boldsymbol{\sigma}_t}{\alpha_t\bar{\boldsymbol{\sigma}}_{t-1}+\boldsymbol{\sigma}_t} = \frac{\sqrt{\bar{\alpha}_{t-1}}(\bar{\beta}_t-\tilde{\beta}_t)g_\psi(\mathbf{X})+\tilde{\beta}_t\boldsymbol{\sigma}_{\mathbf{Y}_0}}{g_\psi(\mathbf{X})(\alpha_t\bar{\beta}_{t-1}-\alpha_t\tilde{\beta}_{t-1}+\beta_t^2)+\boldsymbol{\sigma}_{\mathbf{Y}_0}(\alpha_t\tilde{\beta}_{t-1}+\alpha_t\beta_t)}$$

$$\gamma_2 = \frac{\sqrt{\alpha_t}(\alpha_t-1)\bar{\boldsymbol{\sigma}}_{t-1}+(1-\sqrt{\bar{\alpha}_{t-1}})\boldsymbol{\sigma}_t}{\alpha_t\bar{\boldsymbol{\sigma}}_{t-1}+\boldsymbol{\sigma}_t} \qquad (32)$$

$$= \frac{(\beta_t^2(1-\sqrt{\bar{\alpha}_{t-1}})-\sqrt{\alpha_t}\beta_t(\bar{\beta}_{t-1}-\tilde{\beta}_{t-1}))g_\psi(\mathbf{X})+\alpha_t\beta_t(1-\sqrt{\bar{\alpha}_{t-1}}-\sqrt{\alpha_t}\beta_t\tilde{\beta}_{t-1}))\boldsymbol{\sigma}_{\mathbf{Y}_0}}{g_\psi(\mathbf{X})(\alpha_t\bar{\beta}_{t-1}-\alpha_t\tilde{\beta}_{t-1}+\beta_t^2)+\boldsymbol{\sigma}_{\mathbf{Y}_0}(\alpha_t\tilde{\beta}_{t-1}+\alpha_t\beta_t)}$$

Similarly, for the parameter $\tilde{\sigma}$ in the posteriors of $\mathbf{Y}_{t-1}$, we have:

$$\tilde{\boldsymbol{\sigma}} = \frac{1}{\frac{\alpha_t}{\boldsymbol{\sigma}_t}+\frac{1}{\bar{\boldsymbol{\sigma}}_{t-1}}} = \frac{\boldsymbol{\sigma}_t\bar{\boldsymbol{\sigma}}_{t-1}}{\alpha_t\bar{\boldsymbol{\sigma}}_{t-1}+\boldsymbol{\sigma}_t}$$

$$= \frac{(\beta_t^2 g_\psi(\mathbf{X})+\alpha_t\beta_t\boldsymbol{\sigma}_{\mathbf{Y}_0})((\bar{\beta}_{t-1}-\tilde{\beta}_{t-1})g_\psi(\mathbf{X})+\tilde{\beta}_{t-1}\boldsymbol{\sigma}_{\mathbf{Y}_0})}{g_\psi(\mathbf{X})(\alpha_t\bar{\beta}_{t-1}-\alpha_t\tilde{\beta}_{t-1}+\beta_t^2)+\boldsymbol{\sigma}_{\mathbf{Y}_0}(\alpha_t\tilde{\beta}_{t-1}+\alpha_t\beta_t)} \qquad (33)$$

### A.3. Loss Function

Firstly, KL divergence between two gaussians can be given as:

$$\mathcal{L} = \mathbb{E}\left[D_{\mathrm{KL}}\left(q\left(\mathbf{Y}_{t-1} \mid \mathbf{Y}_t, f_\phi(\mathbf{X}), g_\psi(\mathbf{X})\right) \| p_\theta\left(\mathbf{Y}_{t-1} \mid \mathbf{Y}_t, f_\phi(\mathbf{X}), g_\psi(\mathbf{X})\right)\right)\right]$$

$$= \mathbb{E}\left[D_{\mathrm{KL}}\left(\mathcal{N}(\mathbf{Y}_{t-1};\tilde{\boldsymbol{\mu}},\tilde{\boldsymbol{\sigma}})\|\mathcal{N}\left(\mathbf{Y}_{t-1};\boldsymbol{\mu}_\theta,\boldsymbol{\sigma}_\theta\right)\right)\right]$$

$$= \mathbb{E}\left[\frac{1}{2}\left((\boldsymbol{\mu}_\theta-\tilde{\boldsymbol{\mu}})^\top\boldsymbol{\Sigma}_\theta^{-1}(\boldsymbol{\mu}_\theta-\tilde{\boldsymbol{\mu}})+\mathrm{Tr}\left(\boldsymbol{\Sigma}_\theta^{-1}\tilde{\boldsymbol{\Sigma}}\right)-\log\frac{\det(\tilde{\boldsymbol{\Sigma}})}{\det(\boldsymbol{\Sigma}_\theta)}-C\right)\right] \qquad (34)$$

$$\propto \mathbb{E}\left[\|\boldsymbol{\mu}_\theta-\tilde{\boldsymbol{\mu}}\|_2^2+\sum_i\frac{\tilde{\boldsymbol{\sigma}}_i}{\boldsymbol{\sigma}_{\theta,i}}-\sum_i\log(\frac{\tilde{\boldsymbol{\sigma}}_i}{\boldsymbol{\sigma}_{\theta,i}})\right]$$

where $\Sigma$ is a diagonal matrix representing the covariance matrix and $\sigma$ represents its diagonal vector. Let $\mu_\theta$ be in the form similar to equation 10, and replace $\mathbf{Y}_0$ with $\mathbf{Y}_t$ using Eq. 27, it gives:

$$\boldsymbol{\mu}_\theta = \gamma_0 \left( \underbrace{\frac{1}{\sqrt{\bar{\alpha}_t}} (\mathbf{Y}_t - (1 - \sqrt{\bar{\alpha}_t}) f_\phi(\mathbf{X}) - \sqrt{(\bar{\beta}_{t-1} - \tilde{\beta}_{t-1}) g_\psi(\mathbf{X}) + \tilde{\beta}_{t-1} \boldsymbol{\sigma}_{\mathbf{Y}_0}} \boldsymbol{\eta}_\theta)}_{\mathbf{Y}_0} \right) + \gamma_1 \mathbf{Y}_t + \gamma_2 f_\phi(\mathbf{X}) \qquad (35)$$

Replacing $\tilde{\boldsymbol{\mu}}$ with equation 10 in $\boldsymbol{\mu}_\theta - \tilde{\boldsymbol{\mu}}$, defined as:

$$\boldsymbol{\mu}_\theta - \tilde{\boldsymbol{\mu}} = \gamma_0 \sqrt{(\bar{\beta}_{t-1} - \tilde{\beta}_{t-1}) g_\psi(\mathbf{X}) + \tilde{\beta}_{t-1} \boldsymbol{\sigma}_{\mathbf{Y}_0}} (\boldsymbol{\eta} - \boldsymbol{\eta}_\theta)$$
$$\propto (\boldsymbol{\eta} - \boldsymbol{\eta}_\theta) \qquad (36)$$

Using Eq. 36, the loss function can be derived as:

$$\mathcal{L} = \mathbb{E} \left[ ||\boldsymbol{\eta} - \boldsymbol{\eta}_\theta||_2^2 + \sum_i \frac{\tilde{\boldsymbol{\sigma}}_i}{\boldsymbol{\sigma}_{\theta,i}} - \sum_i \log \left( \frac{\tilde{\boldsymbol{\sigma}}_i}{\boldsymbol{\sigma}_{\theta,i}} \right) \right] \qquad (37)$$

where the first term $||\boldsymbol{\eta} - \boldsymbol{\eta}_\theta||_2^2$ is matching the noise in each step and the remainder $\sum_i \frac{\tilde{\boldsymbol{\sigma}}_i}{\boldsymbol{\sigma}_{\theta,i}} - \sum_i \log \left( \frac{\tilde{\boldsymbol{\sigma}}_i}{\boldsymbol{\sigma}_{\theta,i}} \right)$ is optimizing the uncertainty. Specifically, assume that $\tilde{\boldsymbol{\sigma}} = \boldsymbol{\sigma}_\theta = 1$, which degenerates to the general version of the DDPM: $\mathcal{L} = \mathbb{E} \left[ ||\boldsymbol{\eta} - \boldsymbol{\eta}_\theta||_2^2 \right]$.

### A.4. Estimating $\boldsymbol{\sigma}_{\mathbf{Y}_0}$ Through $\boldsymbol{\sigma}_\theta$

At inference time, the calculation of $\gamma_{0,1,2}$ requires estimating $\boldsymbol{\sigma}_{\mathbf{Y}_0}$. One straightforward approach is to directly use $g(x)$; however, this method assumes a perfect predictor and does not involve the reverse process in parameter estimation. Since the predictor has already provided an estimate of the reverse noise, we utilize the quadratic expansion of equation 33 to approximate $\boldsymbol{\sigma}_{\mathbf{Y}_0}$.

By substituting the predicted $\boldsymbol{\sigma}_\theta$ into equation 33 and rearranging terms, we obtain:

$$\underbrace{\alpha_t \beta_t \tilde{\beta}_{t-1}}_{\lambda_0} \boldsymbol{\sigma}_{\mathbf{Y}_0}^2 + \left( \underbrace{(\beta_t^2 \tilde{\beta}_{t-1} + \alpha_t \beta_t (\bar{\beta}_{t-1} - \tilde{\beta}_{t-1})) g_\psi(\mathbf{X}) - \boldsymbol{\sigma}_\theta (\alpha_t \tilde{\beta}_{t-1} + \alpha_t \beta_t))}_{\lambda_1} \right) \boldsymbol{\sigma}_{\mathbf{Y}_0}$$
$$+ \underbrace{g_\psi(\mathbf{X})^2 \beta_t^2 (\bar{\beta}_{t-1} - \tilde{\beta}_{t-1}) - \boldsymbol{\sigma}_\theta g_\psi(\mathbf{X}) (\alpha_t \bar{\beta}_{t-1} - \alpha_t \tilde{\beta}_{t-1} + \beta_t^2)}_{\lambda_2} = 0 \qquad (38)$$

when $\lambda_2 < 0$, the quadratic equation has exactly one positive root using the Vieta theorem $\frac{\lambda_2}{\lambda_0} > 0$. $\hat{\boldsymbol{\sigma}}_{\mathbf{Y}_0}$ is given by:

$$\hat{\boldsymbol{\sigma}}_{\mathbf{Y}_0} = \frac{-\lambda_1 + \sqrt{\lambda_1^2 - 4\lambda_0 \lambda_2}}{2\lambda_0} \qquad (39)$$

The constraint $\lambda_2 < 0$ can be described by the following inequality:

$$g_\psi(\mathbf{X}) < \boldsymbol{\sigma}_\theta \left( \frac{\alpha_t}{\beta_t^2} + \frac{1}{\bar{\beta}_{t-1} - \tilde{\beta}_{t-1}} \right) \qquad (40)$$

where the coefficient of right-hand side is a very large value; specifically, under our default beta settings with beta start from 0.0001 to 0.02, this value is sufficiently large to ensure the sovlability of Eq. 39; experimentally, the results can be obtained in all datasets and samples.

## A.5. Simplified Versions of NsDiff

### A.5.1. PERFECT ESTIMATOR

Given a perfect estimator $g_\psi(\mathbf{X})$, substituting this into Eq. 6, we obtain the variance $\boldsymbol{\sigma}_t = \beta_t g_\psi(\mathbf{X})$. Substituting this into Eq. 22, Eq. 27 becomes:

$$\mathbf{Y}_t = \sqrt{\bar{\alpha}_t}\mathbf{Y}_0 + \left(1 - \sqrt{\bar{\alpha}_t}\right) f_\phi(\mathbf{X}) + \sqrt{\left(1 - \bar{\alpha}_t\right) g_\psi(\mathbf{X})}\boldsymbol{\eta}_0 \tag{41}$$

hence, we have $\bar{\sigma}_t = \sqrt{\left(1 - \bar{\alpha}_t\right) g_\psi(\mathbf{X})}$. Further follows the deriviation of Appendix A.2 by replacing $\mathbb{A}, \mathbf{B}, \boldsymbol{\sigma}_t$ and $\bar{\boldsymbol{\sigma}}_t$ , we obtain following results:

$$\tilde{\boldsymbol{\mu}} = \frac{\sqrt{\bar{\alpha}_{t-1}}}{1 - \bar{\alpha}_t}\beta_t\mathbf{Y}_0 + \frac{1 - \bar{\alpha}_{t-1}}{1 - \bar{\alpha}_t}\mathbf{Y}_t\sqrt{\alpha_t} + 1 + \frac{\left(\sqrt{\bar{\alpha}_t} - 1\right)\left(\sqrt{\alpha_t} + \sqrt{\bar{\alpha}_{t-1}}\right)}{1 - \bar{\alpha}_t}f_\phi(\mathbf{X}) \tag{42}$$

$$\tilde{\boldsymbol{\sigma}} = \left(1 - \bar{\alpha}_t\right) g_\psi(\mathbf{X}) \tag{43}$$

for the loss function, the posterior variance is known in inference time; hence the training for the variance $\tilde{\sigma}$ is not necessary. This variant can be simply trained using $\mathcal{L} = \mathbb{E}\left[||\boldsymbol{\eta} - \boldsymbol{\eta_\theta}||_2^2\right]$. Furthermore, in this result, the variance is simply a constant scaling of the variance in previous work (Ho et al., 2020; Han et al., 2022; Li et al., 2024a).

### A.5.2. UNIT VARIANCE

Assuming a unit endpoint variance ($g_\psi(\mathbf{X})$=1) gives an identical posterior mean in Eq. 42 and and variance with only the coefficient:

$$\tilde{\boldsymbol{\mu}} = \frac{\sqrt{\bar{\alpha}_{t-1}}}{1 - \bar{\alpha}_t}\beta_t\mathbf{Y}_0 + \frac{1 - \bar{\alpha}_{t-1}}{1 - \bar{\alpha}_t}\mathbf{Y}_t\sqrt{\alpha_t} + 1 + \frac{\left(\sqrt{\bar{\alpha}_t} - 1\right)\left(\sqrt{\alpha_t} + \sqrt{\bar{\alpha}_{t-1}}\right)}{1 - \bar{\alpha}_t}f_\phi(\mathbf{X}) \tag{44}$$

$$\tilde{\boldsymbol{\sigma}} = \left(1 - \bar{\alpha}_t\right) \tag{45}$$

this is also trained in $\mathcal{L} = \mathbb{E}\left[||\boldsymbol{\eta} - \boldsymbol{\eta_\theta}||_2^2\right]$. Note that TMDM (Li et al., 2024a) is a typical work under this setting.

# B. Other Results and Discussions

## B.1. Point Forecast Results

In time series forecasting tasks, mean square error (MSE) and mean average error (MAE) reflect the point estimation accuracy. We provide the results evaluated on these two metrics in Table 6 and Table 7 of the real and syncthetic datasets respectively:

*Table 6.* MAE/MSE result on nine real-world datasets, **bold face** indicate best result.

| Models | Datasets | ETTh1 | ETTh2 | ETTm1 | ETTm2 | ECL | EXG | ILI | Solar | Traffic |
|---|---|---|---|---|---|---|---|---|---|---|
| TimeGrad | MSE | 0.813 | 1.496 | 0.831 | 0.967 | 0.504 | 1.058 | 1.414 | 0.446 | 0.535 |
| (2021) | MAE | 1.062 | 3.462 | 1.218 | 1.690 | 0.505 | 1.567 | 4.197 | 0.475 | 0.983 |
| CSDI | MSE | 0.708 | 0.900 | 0.752 | 1.069 | 0.822 | 1.081 | 1.481 | 0.675 | 0.925 |
| (2022) | MAE | 0.949 | 1.226 | 1.002 | 1.723 | 1.007 | 1.701 | 4.515 | 0.763 | 1.731 |
| TimeDiff | MSE | **0.479** | **0.485** | 0.477 | **0.333** | 0.764 | 0.446 | 1.169 | 0.713 | 0.784 |
| (2023) | MAE | **0.517** | **0.456** | 0.537 | **0.268** | 0.879 | 0.402 | 3.958 | 0.821 | 1.350 |
| DiffusionTS | MSE | 0.774 | 1.411 | 0.744 | 1.232 | 0.856 | 1.564 | 1.788 | 0.740 | 0.815 |
| (2024) | MAE | 1.089 | 3.273 | 1.030 | 2.372 | 1.072 | 3.628 | 6.053 | 0.749 | 1.473 |
| TMDM | MSE | 0.607 | 0.490 | **0.455** | 0.395 | 0.359 | 0.430 | 1.175 | 0.316 | 0.425 |
| (2024) | MAE | 0.696 | 0.512 | 0.494 | 0.315 | 0.257 | 0.334 | 3.636 | 0.250 | 0.679 |
| NsDiff | MSE | 0.523 | 0.490 | **0.455** | 0.352 | **0.306** | **0.412** | **0.985** | **0.307** | **0.373** |
| **(ours)** | MAE | 0.594 | 0.514 | **0.488** | 0.281 | **0.209** | **0.300** | **2.846** | **0.242** | **0.637** |

Note that TimeDiff is a model specifically designed for long-term point forecasting. As in this Table, in datasets with high non-stationarity, NsDiff still achieves SOTA, attributed to the dynamic mean and variance endpoint and the uncertainty-aware noise schedule.

*Table 7.* MAE/MSE result on two synthetic datasets, **bold face** indicate best result.

| Variance | Linear | | Quadratic | |
|---|---|---|---|---|
| Models | MSE | MAE | MSE | MAE |
| TimeGrad | 1.546 | 3.726 | 2.629 | 10.677 |
| CSDI | 1.516 | 3.641 | 2.553 | 10.204 |
| TimeDiff | 1.537 | 3.776 | 2.649 | 11.275 |
| DiffusionTS | 1.738 | 4.766 | 2.504 | 9.620 |
| TMDM | 1.514 | 3.639 | 2.582 | 10.403 |
| NsDiff | **1.512** | **3.616** | **2.490** | **9.543** |

On the synthetic dataset, the advantages of NsDiff are more pronounced, due to the high variation of both mean and variance in the dataset. This proves NsDiff's performance under non-stationary environment.

### B.2. Effects of Pretraining

Although NsDiff follows the pretraining paradigm to stabilize training and achieve optimal performance, end-to-end training remains a viable alternative. Our experiments demonstrate that the networks can be trained jointly without significant performance degradation. For instance, when evaluating on the ETTh1 dataset with and without pretraining, we observe comparable results in the CRPS metric. We report the results in Table 8.

*Table 8.* The comparison between pretraining and end-to-end training, **bold face** indicate best result.

| epoch | pretrain | end-to-end |
|---|---|---|
| 1 | 0.4181 | 0.4407 |
| 2 | 0.4041 | 0.4227 |
| 3 | 0.3977 | 0.4045 |
| 4 | 0.3926 | 0.4004 |
| 5 | 0.3889 | **0.3868** |
| 6 | **0.3795** | 0.3873 |

As can be seen, although joint train experiences a slight performance degradation (1.86%), it still outperforms the previous state-of-the-art TMDM (0.452). However, compared to pretraining, co-training is slightly harder to converge.

### B.3. Computation Efficiency

To analyse the computational efficiency of NsDiff, we compare the training and inference memory and time cost and performance, and report the results in Table 9.

*Table 9.* Computation efficiency comparison, **bold face** indicate best result.

| Model | Mem.Train(MB) | Mem.Inference(MB) | Tim.Train(ms) | Tim.Inference(ms) | CRPS | QICE |
|---|---|---|---|---|---|---|
| TimeGrad | 27.47 | 8.61 | 47.89 | 8319.29 | 0.606 | 6.731 |
| CSDI | 109.81 | 22.61 | 60.50 | 446.70 | 0.492 | 3.107 |
| TimeDiff | **15.66** | **3.40** | 33.93 | 238.78 | 0.465 | 14.931 |
| DiffusionTS | 65.03 | 79.23 | 94.51 | 8214.53 | 0.603 | 6.423 |
| TMDM | 221.58 | 213.46 | 33.26 | 237.37 | 0.452 | 2.821 |
| NsDiff | 68.20 | 57.75 | **32.13** | **208.07** | **0.392** | **1.470** |

As shown in Table 9, compared to previous SOTA TMDM, NsDiff achieves SOTA and has smaller memory costs and higher

efficiency. This is because NsDiff does not introduce additional hidden variables and only adds a small number of basic operations. It should be noted that while NsDiff does not achieve the lowest memory cost, primarily due to its default use of the relatively heavy Non-stationary Transformer mean estimator (Liu et al., 2022), the NsDiff framework itself is model-agnostic. As such, the mean estimator can be replaced with a lighter conditional expectation forecasting model e.g. DLinear (Zeng et al., 2023) to optimize memory efficiency.

# C. Reproducibility

We provide all relevant data, code, and notebooks at `https://github.com/wwy155/NsDiff`.

## C.1. Datasets

### C.1.1. REAL DATASET

Nine real-world datasets with varying levels of uncertainty were chosen, including: (1) Electricity[1] - which documents the hourly electricity usage of 321 customers from 2012 to 2014. (2) ILI[2] - which tracks the weekly proportion of influenza-like illness (ILI) patients relative to the total number of patients, as reported by the U.S. Centers for Disease Control and Prevention from 2002 to 2021. (3) ETT (Zhou et al., 2021) - which includes data from electricity transformers, such as load and oil temperature, recorded every 15 minutes between July 2016 and July 2018. (4) Exchang (Lai et al., 2018) - which logs the daily exchange rates of eight countries from 1990 to 2016. (5) Traffic[3] - which provides hourly road occupancy rates measured by 862 sensors on San Francisco Bay area freeways from January 2015 to December 2016. (6) SolarEnergy[4] - a dataset from the National Renewable Energy Laboratory containing solar power output data collected from 137 photovoltaic plants in Alabama in 2007.

### C.1.2. SYNTHETIC DATASETS

To accurately evaluate the performance of NsDiff under time-varying conditions, we design two synthetic datasets using the LSNM. Specifically, the data generation follows the formula:

$$\mathbf{Y} = \mathbf{m}[\mathbf{t}] + \mathbf{v}[\mathbf{t}]\epsilon,$$

where $\mathbf{m}$ and $\mathbf{v}$ define the trend level and the uncertainty variation, respectively. The results of these experiments are summarized in Table 4. We provide the generation codes below:

**Linear Synthetic Dataset:**

```python
import numpy as np

def generate_synthetic_data(length):
    means = np.linspace(1, 10, length)   # means from 1 to 10
    stddev = np.linspace(1, 10, length)   # standard deviations from 1 to 10

    data = np.zeros(length)
    for t in range(length):
        data[t] = np.random.normal(loc=means[t], scale=stddev[t])
    return data
```

**Quadratic Synthetic Dataset:**

```python
import numpy as np

def generate_synthetic_data(length):
    means = np.linspace(1, 10, length)   # means from 1 to 10
```

---

[1]`https://archive.ics.uci.edu/ml/datasets/ElectricityLoadDiagrams20112014`
[2]`https://gis.cdc.gov/grasp/fluview/fluportaldashboard.html`
[3]`http://pems.dot.ca.gov/`
[4]`http://www.nrel.gov/grid/solar-power-data.html`

```
    stddev = np.linspace(1, 10, length)   # standard deviations from 1 to 10

    data = np.zeros(length)
    for t in range(length):
        data[t] = np.random.normal(loc=means[t], scale=stddev[t]*stddev[t])
    return data
```

In the linear setting, $\mathbf{m}$ increases linearly from 1 to 10, and $\mathbf{v}$ follows the same pattern. In contrast, for the quadratic setting, $\mathbf{v}$ grows quadratically from 1 to 100. The total length of the generated dataset is 7588, and we predict univariate feature.

### C.2. $g_\psi(\mathbf{X})$ implementation

#### C.2.1. COMPUTE $\sigma_{\mathbf{Y}_0}$

The ground truth variance can be estimated in various ways, such as using specific dates or a sliding window. Given the proven success of employing sliding windows in time series analysis to predict variance (Liu et al., 2024b), we adopt the sliding window approach to extract the ground truth variance. The Python code for computing $\sigma_{\mathbf{Y}_0}$ is provided below.

```
def y_sigma(x, y, window_size=96):
    """
    Compute variance using a sliding window.

    Args:
        x (torch.Tensor): Input tensor (B, T, N).
        y (torch.Tensor): Output tensor (B, O, N).
        window_size (int): Sliding window size (default: 96).

    Returns:
        torch.Tensor: Variance tensor (B, O, N).
    """
    all_data = torch.cat([x, y], dim=1)   # Combine input and output
    windows = all_data.unfold(1, window_size, 1)   # Create sliding windows
    sigma = windows.var(dim=3, unbiased=False)   # Compute variance
    return sigma[:, -y.shape[1]:, :]   # Extract output step variance
```

#### C.2.2. ARCHETECUTURE

The architecture of a pretrained variance estimator can take various forms. Without loss of generality, we employ a simple 3-layer Multi-Layer Perceptron (MLP) as the variance estimator. The MLP is configured with a hidden size of 512 and utilizes ReLU activations between the layers. The PyTorch implementation of this architecture is provided below:

```
nn.Sequential(
    nn.Linear(seq_len, hidden_size),
    nn.ReLU(),
    nn.Linear(hidden_size, hidden_size),
    nn.ReLU(),
    nn.Linear(hidden_size, pred_len)
)
```

### C.3. Metrics

**CRPS:** The continuous ranked probability score (CRPS) (Matheson & Winkler, 1976) measures the compatibility of a cumulative distribution function (CDF) $F$ with an observation $x$ as

$$\text{CRPS}(F, x) = \int_{\mathbb{R}} (F(z) - \mathbb{I}\{x \leq z\})^2 \, \mathrm{d}z \tag{46}$$

where $\mathbb{I}_{z<q}$ is an indicator function. Employing the empirical CDF of $F$, i.e. $\hat{F}(z) = \frac{1}{S} \sum_{s=1}^{S} \mathbb{I}\left\{x^{0,s} \leq z\right\}$ with $S$ samples $x^{0,s} \sim F$ as a natural approximation of the predictive CDF, CRPS can be directly computed by samples from DDPM. We generated 100 samples to approximate the distribution $F$.

**QICE:** The quantile interval calibration error (QICE) (Han et al., 2022) quantifies the deviation between the proportion of true data contained within each quantile interval (QI) and the optimal proportion, which is $1/M$ for all intervals. To compute QICE, we divide the generated $y$-samples into $M$ quantile intervals with roughly equal sizes, corresponding to the boundaries of the estimated quantiles. Under the optimal scenario, when the learned distribution matches the true distribution, each QI should contain approximately $1/M$ of the true data. QICE is formally defined as the mean absolute error between the observed and optimal proportions, and can be expressed as:

$$\text{QICE} := \frac{1}{M} \sum_{m=1}^{M} \left| r_m - \frac{1}{M} \right|, \tag{47}$$

where $r_m = \frac{I}{N} \sum_{n=1}^{N} \mathbb{I}_{y_n \geq \hat{y}_n^{\text{low}_m}} \cdot \mathbb{I}_{y_n \leq \hat{y}_n^{\text{high}_m}}$. Here, $\mathbb{I}_{\text{condition}}$ is an indicator function. The terms $\hat{y}_n^{\text{low}_m}$ and $\hat{y}_n^{\text{high}_m}$ denote the lower and upper boundaries of the $m$-th quantile interval, respectively. Intuitively, under ideal conditions with sufficient samples, QICE should approach 0, indicating that each QI contains the expected proportion of data. Following Li et al. (2024a), we calculate the QICE by partitioning the probability range into ten equal decile-based intervals.

## D. ShowCases

We present additional results in Figure 5, which clearly demonstrate that NsDiff effectively captures the inherent uncertainty in the data, even in the presence of significant variations. Notably, in the ILI dataset, NsDiff accurately identifies the reduced variance, a feature that other methods fail to detect. Moreover, on the ExchangeRate dataset—a highly volatile financial dataset—NsDiff successfully identifies both the substantial variance and the overall trend, providing precise interval estimates. In contrast, other methods exhibit notable shortcomings: for instance, TimeGrad predicts an excessively large downward trend, CSDI produces overly wide intervals, and TMDM fails to adequately cover the data range. These results underscore the robustness and accuracy of NsDiff in handling diverse and challenging datasets.

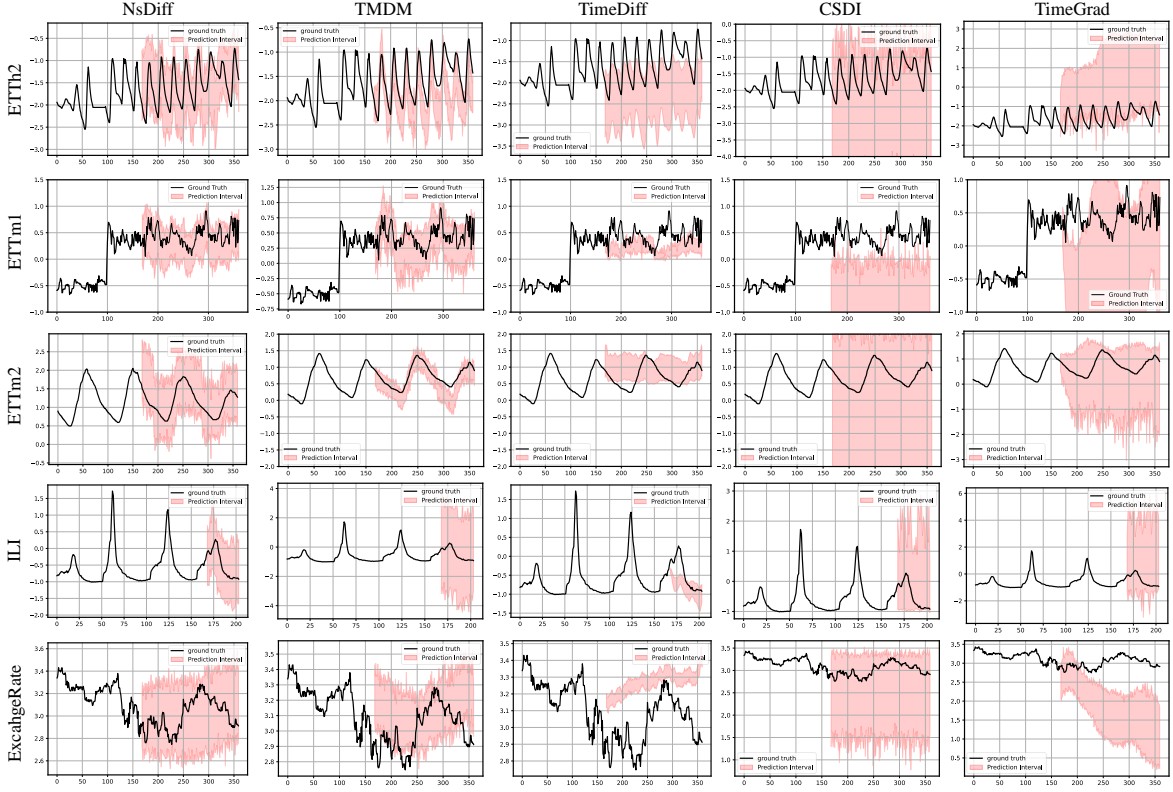

*Figure 5.* The 95% prediction intervals comparison with other models.

