# OpenReview forum: "Non-stationary Diffusion For Probabilistic Time Series Forecasting"
_ICML.cc/2025/Conference — ICML 2025 spotlightposter_

### Official Review · Reviewer_ZzLi · 2025-02-22

**Overall Recommendation:** 4

**Summary:**

This paper introduces NsDiff, a novel diffusion-based framework for probabilistic time series forecasting that explicitly addresses non-stationary uncertainty. Recognizing that conventional DDPMs rely on a fixed variance assumption from the additive noise model (ANM), the authors propose the integration of a Location-Scale Noise Model (LSNM) to allow the variance to vary with the input data. NsDiff combines a pre-trained conditional mean and variance estimator with an uncertainty-aware noise schedule that dynamically adapts noise levels at each diffusion step. Extensive experiments on nine real-world and synthetic datasets demonstrate that NsDiff significantly outperforms existing methods, such as TimeGrad and TMDM, especially in capturing changing uncertainty patterns. Although the paper provides thorough theoretical derivations and promising empirical results, some aspects—such as the integration of the pre-trained estimators and the robustness of the noise schedule—could benefit from further clarification.

1. Why does NsDiff not adopt a fully end-to-end joint optimization approach? Is it necessary to pre-train the two networks separately?

2. Figure 2 contains numerous curved lines. While I understand that this may reflect the authors’ intentional design style, it appears rather unappealing and should be modified.

3. Although the task focuses on probabilistic MTS forecasting, I recommend additionally reporting metrics such as MSE and MAE, since the mean is an important characteristic of the distribution.

**Claims And Evidence:**

See the summary.

**Essential References Not Discussed:**

See the summary.

**Experimental Designs Or Analyses:**

See the summary.

**Methods And Evaluation Criteria:**

See the summary.

**Other Comments Or Suggestions:**

See the summary.

**Other Strengths And Weaknesses:**

See the summary.

**Questions For Authors:**

See the summary.

**Relation To Broader Scientific Literature:**

See the summary.

**Theoretical Claims:**

See the summary.

---

> ### Author Rebuttal · Authors · 2025-03-31
>
> **Q1**: Why does NsDiff not adopt a fully end-to-end joint optimization approach? Is it necessary to pre-train the two networks separately?
>
> **A1**: yes, the networks can be trained jointly without large performance loss. Below is an example to train on ETTh1 with and without pretraining, where we report CRPS metric.
>
> |epoch|pretrain|jointtrain|
> |-------|----------|----------|
> |1|0.4181|0.4407|
> |2|0.4041|0.4227|
> |3|0.3977|0.4045|
> |4|0.3926|0.4004|
> |5|0.3889|**0.3868**|
> |6|**0.3795**|0.3873|
> |MSE earlystop|||
>
> As can be seen, although joint train experiences a slight performance degradation (1.86%), it still outperforms the previous state-of-the-art TMDM (0.452). However, compared to pretraining, co-training is slightly harder to converge. We will clarify this in the updated version.
>
> **Q2**: Figure 2 should be modified, e.g. remove curved lines.
>
> **A2**: Thanks for this suggestion, we will modify Figure 2 according to your advice in the lastest version.
>
> **Q3**: Report additional metrics such as MSE/MAE.
>
> **A3**: Thanks for your insight. We give the results on real and synthetic datasets at the following tables with an addtional baseline CSBI (as required by Reviewer gSNF), the settings are consistent with the main context. The repo is updated accordingly to include CSBI code.
>
> We present the results of MSE/MAE of synthetic datasets at follows:
> |Variance|Linear||Quadratic||
> |-----------|------|---------|---------|---------|
> |Models|MSE|MAE|MSE|MAE|
> |TimeGrad|1.546|3.726|1.626|4.173|
> |CSDI|1.516|3.641|1.546|3.768|
> |TimeDiff|1.537|3.776|1.559|3.762|
> |DiffusionTS|1.738|4.766|1.689|4.823|
> |TMDM|1.514|3.639|1.493|3.568|
> |NsDiff|**1.512**|**3.616**|**1.479**|**3.448**|
>
> As seen in this Table, NsDiff still achieves SOTA in uncertainty variation conditions.
>
> We present the results of MSE/MAE of real datasets at follows:
>
> |Models|Datasets|ETTh1|ETTh2|ETTm1|ETTm2|ECL|EXG|ILI|Solar|Traffic|
> |-------------|----------|--------|--------|--------|--------|--------|--------|--------|--------|---------|
> |TimeGrad|MSE|0.813|1.496|0.831|0.967|0.504|1.058|1.414|0.446|0.535|
> |(2021)|MAE|1.062|3.462|1.218|1.690|0.505|1.567|4.197|0.475|0.983|
> |CSDI|MSE|0.708|0.900|0.752|1.069|0.822|1.081|1.481|0.675|0.925|
> |(2022)|MAE|0.949|1.226|1.002|1.723|1.007|1.701|4.515|0.763|1.731|
> |CSBI|MSE|0.634|0.820|0.757|0.636|0.783|0.897|1.438|0.651|0.848|
> |(2023)|MAE|0.762|0.659|0.526|0.841|0.923|0.746|4.344|0.748|1.527|
> |TimeDiff|MSE|**0.479**|**0.485**|0.477|**0.333**|0.764|0.446|1.169|0.713|0.784|
> |(2023)|MAE|**0.517**|**0.456**|0.537|**0.268**|0.879|0.402|3.958|0.821|1.350|
> |DiffusionTS|MSE|0.774|1.411|0.744|1.232|0.856|1.564|1.788|0.740|0.815|
> |(2024)|MAE|1.089|3.273|1.030|2.372|1.072|3.628|6.053|0.749|1.473|
> |TMDM|MSE|0.607|0.490|**0.455**|0.395|0.359|0.430|1.175|0.316|0.425|
> |(2024)|MAE|0.696|0.512|0.494|0.315|0.257|0.334|3.636|0.250|0.679|
> |NsDiff|MSE|0.523|0.490|**0.455**|0.352|**0.306**|**0.412**|**0.985**|**0.307**|**0.373**|
> |(ours)|MAE|0.594|0.514|**0.488**|0.281|**0.209**|**0.300**|**2.846**|**0.242**|**0.637**|
>
> Note: TimeDiff is a model speciffically designed for long-term point forecasting.
>
> As in this Table, in datasets with high non-stationarity, NsDiff still achieves SOTA, attributed to the dynamic mean and variance endpoint and the uncertainty-aware noise schedule.
>
>
> Here we give the results of an addtional baseline CSBI, where NsDiff still remains SOTA.
> |Models|Datasets|ETTh1|ETTh2|ETTm1|ETTm2|ECL|EXG|ILI|Solar|Traffic|
> |-------------|----------|---------|---------|---------|---------|---------|---------|---------|---------|---------|
> |CSBI|CRPS|0.552|0.571|0.502|0.491|0.585|0.659|1.109|0.498|0.875|
> |(2023)|QICE|6.141|5.230|3.471|8.918|7.982|6.870|7.175|10.830|11.382|

---

### Official Review · Reviewer_gSNF · 2025-03-08

**Overall Recommendation:** 3

**Summary:**

This paper introduces a new probabilistic time series forecasting method based on non-stationary diffusion by estimation the step-wise means and variances. The proposed method is validated on different real-world datasets.

**Claims And Evidence:**

No. Below are some of my concerns.
1)Estimating the variance of time series is numerically tricky. Please clarify how to predict the variances in a numerically stable way using pretraining in Sec.4.3. And what if the variance predicted via MLE is too large (when some spiky data points appear)? Using sliding windows might not address this issue perfectly.

2)From Algorithm 1, it seems that the proposed training method is in fact based on fine-tuning, which requires path sampling during training. Is it too computationally expensive for diffusion-based time series forecasting models?

**Essential References Not Discussed:**

No.

**Experimental Designs Or Analyses:**

Yes.

1)The CRPS and QICE may not be sufficient to evaluate the model’s forecasting performance.

2)The authors may also need to report the time efficiency and memory costs of the proposed training and inference procedures.

**Methods And Evaluation Criteria:**

No.
Apart from the results reported in Table 3, it would be more convincing if the authors could report the relevant RMSEs and MAEs to show the proposed method is an unbiased point-wise predictor. In my opinion, PICP and QICE may be more suitable for evaluating uncertainty quantification tasks than forecasting tasks.

**Other Comments Or Suggestions:**

Please include a detailed account of how the proposed method builds upon and differs from prior works, e.g., [1]., with appropriate citations; in the Appendix and main text.

**Other Strengths And Weaknesses:**

Strengths:
1) The writing is clear and easy to follow.

2) Including uncertainty estimation for time series diffusion model is reasonable.

3) The relevant arithmetic prof regarding the computation of the step-wise variances is given.

Weaknesses:
1) The novelty of the proposed method is limited as it heavily builds upon the existing method [1].

2) The numerically stability of the proposed method is somewhat questionable.

3) The computational cost of proposed method is high compared to existing diffusion methods, e.g., CSDI.

4)Some important evaluation metrics for forecasting are missing, e.g., MAE and RMSE.

**Questions For Authors:**

Given the substantial computational cost of training this proposed method compared to other diffusion-based models, is the performance improvement significant enough to justify the expense?

**Relation To Broader Scientific Literature:**

The proposed method is largely built on the following paper.

Reference:
[1] Li, Y., Chen, W., Hu, X., Chen, B., Zhou, M., et al. Transformer-modulated diffusion models for probabilistic multivariate time series forecasting. In The Twelfth International Conference on Learning Representations, 2024.

The derivations in Appendix A in this paper (which is the core of the proposed method) is directly adapted from Appendix D in [1].

**Theoretical Claims:**

Yes.

---

> ### Author Rebuttal · Authors · 2025-03-31
>
> **Q1**: The differences from TMDM.
>
> **A1**: We believe there are some misunderstandings regarding the relationship between our method and TMDM, particularly in the contributions and derivation aspects. We believe the differences from TMDM are clearly presented throughout the paper. To aid clarity, we list some key differences between NsDiff and TMDM along with parts of where they are discussed in the paper. We hope the following table helps the reviewer quickly identify and understand these distinctions with TMDM.
>
> | differences| mentioned in|
> | -- | -- |
> | TMDM uses ANM assumption, while NsDiff uses LSNM assumption. | Figure 1, right to line 23-27.|
> | TMDM uses $\mathcal{N}(f_\phi(x),\mathbf{I})$ endpoint while NsDiff uses $\mathcal{N}(f_\phi(x),g_\psi(x))$ endpoint | Figure 1, left to line 57-74. |
> | TMDM uses traditional noise schedule, while NsDiff introduces uncertainty-aware noise schedule.| Section 4.6 gives how NsDiff can degenerated to TMDM. Table 5 further gives exp. results of different noise schedule.|
> | TMDM does not optimize variance in loss function | Eq. 13 where TMDM has only the left term. |
> | TMDM does not estimate reverse variance. | line 284 in Algorithms 2. |
> | TMDM could not handle non-stationary variance | Table 3/Table 4 gives the exp. results on real/synthetic dataset, Figure 3/4 gives a visualized illustration.|
>
> We believe that in the derivation presented in Appendix A, all the relevant parts listed in the table above are different from those in TMDM. For example, NsDiff infer the reverse distribution variance by solving Eq. 38, while TMDM does not introduce this step.
>
> **Q2**: the method is based on fine-tuning, which requires path sampling during training. Is this computationally expensive for DDPM in time series?
>
> **A2**: The fact is **no path sampling is required during training**. The training and inference procedures are the same as those of a standard DDPM, except for using a different endpoint and noise schedule. We only introduce some basic operations during training and inference, so the overall computational complexity remains. See **Q5** for experimental results.
>
> **Q3**: What if the variance predicted via MLE is too large (when some spiky data points appear)? Using sliding windows might not address this issue perfectly.
>
> **A3**: We do introduce a design to address this issue by using an uncertainty-aware noise schedule, which incorporates the true variance into the diffusion training process. This design reduces reliance on the variance predicted by the estimator (e.g., via MLE), which can be overly large in the presence of spiky or noisy data. See Table 5 for the experiments results, where it can be seen that by explicitly learning from the true variance, NsDiff becomes more robust to such cases, rather than relying solely on sliding window heuristics or estimator outputs.
>
>
> **Q4**: The numerically stability of NsDiff.
>
> **A4**: In NsDiff, the only potential numerical issue is the solvability of the equation in Eq. 15. We have already provided the conditions for solvability in Eq. 17. As stated in the paper (lines 269-272), this equation always has a solution. Therefore, theoretically, **NsDiff does not introduce additional numerical instability issues. In practice, no numerical instability has been observed in our experiments.** We kindly invite reviewers to check and run our code to verify this.
>
> **Q5**: The efficiency and memory costs in training and inference phases vs. performance improvements.
>
> **A5**: As shown in the following table, compared to TMDM, NsDiff achieves SOTA and has smaller memory costs and higher efficiency. This is because NsDiff does not introduce additional hidden variables and only adds a small number of basic operations.
>
>
> |Model|Mem.Train(MB)|Mem.Inference(MB)|Tim.Train(ms)|Tim.Inference(ms)|CRPS|QICE|
> |----|---|--|---|--|--|--|
> |TimeGrad|27.47|8.61|47.89|8319.29|0.606|6.731|
> |CSDI|109.81|22.61|60.50|446.70|0.492|3.107|
> |TimeDiff|**15.66**|**3.40**|33.93|238.78|0.465|14.931|
> |DiffusionTS|65.03|79.23|94.51|8214.53|0.603|6.423|
> |TMDM|221.58|213.46|33.26|237.37|0.452|2.821|
> |NsDiff|68.20|57.75|**32.13**|**208.07**|**0.392**|**1.470**|
>
> The results are tested on ETTh1, with 100 diffusion steps.
>
> **Q6**: Some metrics are missing, e.g., MAE.
>
> **A6**: NsDiff still achieves SOTA on these metrics, see **Reviewer ZzLi Q3**.

---

> > ### Comment · Reviewer_gSNF · 2025-04-03
> >
> > Thanks for the detailed response. My recommendation score has been updated.

---

### Official Review · Reviewer_iEvE · 2025-03-09

**Overall Recommendation:** 4

**Summary:**

In this paper, the authors considered modeling the uncertainty quantification when applying diffusion models to time-series forecasting tasks. In the beginning, the authors first demonstrated a toy case study that the DDPM may not perform well on uncertainty prediction tasks due to the traditional Additive Noise Model (ANM) scheme. After that, the authors designed the location scale noise model to alleviate this issue, and proposed the Non-stationary Diffusion Model (NsDiff) framework. In the NsDiff, the authors redesigned the forward noise process and reformulated the backward generation process rigorously. Finally, the authors conducted various experiments to demonstrate the efficacy of the proposed approach.

**Claims And Evidence:**

The claims made in this submission are supported by rigorous and convincing evidence. However the reviewer has the following two issues:
1. On page 3, Eq. 1. Given that diffusion models' inference is solving the SDE/ODE, should this part be given as the integral form?
2. In Figure 2, the proposed paper, to the reviewer's understanding mainly focuses on the uncertainty prediction of time series. However, the picture during the inference stage, does not include the uncertainty interval.

**Essential References Not Discussed:**

As mentioned above, there remain the related works from the following two aspects have not been discussed:
1. Bridge-based Models: The diffusion models designed by diffusion bridges [1,2] have not been well discussed. The DDPM can be treated by a special kind of Ornstein–Uhlenbeck bridge [3].
2. The noise editing: It seems that the noise was modified to delineate the predicted time-series result. Thus, related works on noise selection [4] could be considered to some extent.

---
References:
[1]. Provably Convergent Schrodinger Bridge with Applications to Probabilistic Time Series Imputation, ICML-2023
[2]. Flow Matching for Generative Modeling, ICLR 2023
[3]. Image Restoration Through Generalized Ornstein-Uhlenbeck Bridge, ICML 2024
[4] Xiefan Guo etal, Initno: Boosting text-to-image diffusion models via initial noise optimization. CVPR, 2024

**Experimental Designs Or Analyses:**

1. In the supplementary material, Figure 5, the TMDM model can perform well under ETT datasets compared to the proposed NsDiff, what causes this result?
2. The computational time has not been proposed.
3. Is it possible to apply the diffusion model solvers like DPM solver during the model inference stage?
4. As mentioned above, the baseline comparison lacks related baseline models like Schrodinger bridge imputation models.
---
References:
[1]. DPM-Solver: A Fast ODE Solver for Diffusion Probabilistic Model Sampling in Around 10 Steps, NeurIPS 2022

**Methods And Evaluation Criteria:**

Yes, the methods and evaluation criteria make sense for demonstrating the derivation of convergence analysis. Nevertheless, the reviewer still has the following concerns:
1. To the best of the reviewer's knowledge, the authors attempt to transform some predicted value (distribution 1) to another similar distribution, which includes the uncertainty information. Based on this, to the reviewer's understanding, this problem can be treated as a kind of Schrodinger bridge problem. In addition, related works have applied the Schrodinger bridge [1] in the imputation procedure, which is similar to the CSDI model.
2. Regarding the evaluation criteria, to the best of the reviewer's knowledge, time-series forecasting mainly focuses on prediction accuracy. The authors have not included related evaluation metrics like mean square error or mean absolute error [2]. It would be better to clarify this issue.

---
References:
[1]. Provably Convergent Schrodinger Bridge with Applications to Probabilistic Time Series Imputation, ICML-2023
[2]. Transformers in Time Series: A Survey

**Other Comments Or Suggestions:**

See the abovementioned chat window.

**Other Strengths And Weaknesses:**

### Strengths
1. The topic is related to the ICML conferences.
2. The proposed approach is interesting.
3. The derivation is rigorous.

### Weaknesses
1. Major weaknesses have been listed in the abovementioned items.
2. To the reviewer's knowledge, the initial value is of great importance for solving ODE. What would happen if the pre-trained model does not predict well (in the context of ODE, we call it stiffness [1])? It would be better to demonstrate the results under various random seeds to demonstrate the robustness of the proposed approach.
3. The convergence of the proposed approach has not been discussed.
---
References:
[1]. Numerical Methods for Ordinary Differential Equations

**Questions For Authors:**

See the abovementioned chat window.

**Relation To Broader Scientific Literature:**

The diffusion model is of great importance and applying the diffusion model to model the uncertainty information in the context of time-series forecasting is of great necessity. Thus, the key contributions of the paper related to the broader scientific literature are of great important.

**Theoretical Claims:**

The reviewer attempts to check the derivation of the theorem and it seems that nearly all the derivations look good to the best of the reviewer's knowledge.

---

> ### Author Rebuttal · Authors · 2025-03-31
>
> **Q1**: should Eq. 1. be given as the ODE/SDE integral form?
>
> **A1**: Thanks for the insight. However, we believe the reviewer may be referring to Eq. 7 instead of Eq. 1: Eq. 1 describes the LSNM and is not a stochastic process, so it cannot be written as SDE; Eq. 7 defines the data perturbation process and can be written as SDE. We give a theoretical discussion as follows.
> First, the Euler-Maruyama discrete form of NsDiff is
>
>
> $$
> \mathbf{Y}(t+\triangle t) = Y(t) -\frac{1}{2}\beta(t)(\mathbf{Y}(t) - f_\phi(\mathbf{X}))\triangle t + \sqrt{\sigma_{Y_0} - \beta(t)(\sigma_{Y_0} -g_\psi(\mathbf{X}))\triangle t} \sqrt{\beta(t)\triangle t}\mathbf{z}(t)
> $$
>
> The diffusion coefficient is related to the $\triangle t$, which is non-Itô-integrable, making it intractable to define a clean continuous-time reverse SDE for the process. To give more theoretical insight, we resort to the perfect estimator assumption (i.e., $\sigma_{Y_0}=g_\psi(\mathbf{X})$) and give the forward/reverse SDE as follows:
>
> $$
> d\mathbf{Y} = -\frac{1}{2}\beta(t)(\mathbf{Y} - f_\phi(\mathbf{X}))dt + \sqrt{g_\psi(\mathbf{X})\beta(t)}d\mathbf{w}
> $$
>
> $$
> d\mathbf{Y} = [-\frac{1}{2}\beta(t)(\mathbf{Y} - f_\phi(\mathbf{X}))-g_\psi(\mathbf{X})\beta(t)\nabla_\mathbf{Y}\log p_t(\mathbf{Y})]dt + \sqrt{g_\psi(\mathbf{X})\beta(t)}d\mathbf{\bar w}
> $$
>
> We will include more detail in the latest version.
>
> **Q2**: improve Figure 2 and the description of Eq. 21.
>
> **A2**: Thanks for this suggestion, we will revise the paper according to your advice.
>
> **Q3**: SB problem and include SB-based baselines.
>
> **A3**: Indeed, the problem can be viewed as a SB problem. Taking CSBI as an example, we can introduce an uncertainty-aware prior for it by replacing its $p_\text{prior}$ with the prior of NsDiff $\mathcal{N}(f_\phi(X),g_\psi(X))$, to combine the advantages of NsDiff. Compared to CSBI, NsDiff can estimate the true variance $\sigma_{Y_0}$ by $\sigma_\theta$ (see App. A.4)  to learn uncertainty more accurately (as reflected in the results of Sec. 5.4). We include a SB-based baseline CSBI in Table 3 and 4, see **Revewer ZzLi Q3**.
>
> **Q4**: MSE/MAE, computational time.
>
> **Q4**: NsDiff still achieves SOTA on MSE/MAE. Furthermore, compared to previous SOTA, NsDiff has smaller memory costs and higher efficiency. This is because NsDiff does not introduce additional hidden variables and only adds a small number of basic operations.
>
> **Q5**: Figure 5, the TMDM model seems to perform better, why?
>
> **A5**: In Figure 5, **TMDM actually performs worse than NsDiff**. Of course, the difference is less significant compared to Traffic as ETT datasets have relatively small uncertainty variation as in Table 2 (e.g. 1.2 ETTm1, 1.3 ETTm2). Specifically, Figure 5 shows TMDM produces a less accurate mean prediction, leading to larger MAE and MSE than Nsdiff; and,  TMDM's predictions do not sufficiently cover the true values, resulting in poorer CRPS and QICE scores.
>
> **Q6**: Is it possible to apply the diffusion model solvers during inference stage?
>
> **A6**: Yes, following A1, NsDiff has the following ODE form:
> $$
> d\mathbf{Y} = [-\frac{1}{2}\beta(t)(\mathbf{Y} - f_\phi(\mathbf{X}))- \frac{1}{2}g_\psi(\mathbf{X})\beta(t)\nabla_\mathbf{Y}\log p_t(\mathbf{Y})]dt
> $$
> where the ODE follows a semi-linear structure, remaining compatible with DPM-Solver. However, since time series tasks typically require only a few steps (less than 100) for effective performance, the inference efficiency is already sufficient. Therefore, we do not recommend applying DPM-Solver at the cost of prediction accuracy, given an additional assumption in A1.
>
> **Q7**: Bridge-based Models [1-3]  and noise editing [4]  should be discussed.
>
> **A7**: Thanks for these references. we have discussed SB-based methods in Q3. The noising editing method is interesting and relevant. While NsDiff gives a distribution-wise improvement on the initial noise, the given paper consider a sample-wise perspective to optimize the initial noise to a more reasonable space. We will provide more discussion in the latest version.
>
> **Q8**: What would happen if the pre-trained model does not predict well (stiffness). Include various random seeds to demonstrate the robustness.
>
> **A8**: NsDiff incorporates the true variance $\sigma_{Y_0}$ into the learning process to alleviate this stiffness problem. Although the pre-trained model $g_\psi(X)$ may not predict well, NsDiff use Eq. 18 to estimate the true variance. As shown in Table 5 of the ablation experiments (we report results mean and std on various seeds), incorporating this variance estimation improves both the performance and the robustness of our method.
>
> **Q9**: The convergence of NsDiff has not been discussed.
>
> **A9**: We acknowledge that the convergence analysis is a critical problem, which is missing in basically all previous works, e.g. TimeGrad, TMDM etc. We will explore this more in future work.

---

> > ### Comment · Reviewer_iEvE · 2025-04-05
> >
> > The reviewer appreciates the authors' detailed and thoughtful rebuttal. However, the reviewer would like to suggest a few additional revisions to further strengthen the rigor and clarity of the work:
> > 1. It would enhance the rigor of the manuscript to include a demonstration of convergence results across epochs.
> > 2. Based on the descriptions of Algorithm 1 and Algorithm 2, it seems beneficial to summarize them into a combined Algorithm 3. This would help rectify and streamline the workflow of the proposed approach.
> > 3. Since Table 5 presents experiments conducted with various seeds, it would be more robust to include a paired-sample $t$-test to statistically validate the results.

---

> > > ### Author Response · Authors · 2025-04-06
> > >
> > > **Q10**: a demonstration of convergence results across epochs.
> > >
> > > **A10**: According to your advice, we provide a figure visual illustration with train loss/test result across epochs of  $f_\phi(x)$, $g_\psi(x)$, NsDiff at https://1drv.ms/i/c/f104f0574e8cb377/EQz0OZHb9ahKtv2ZU9tA5HIBVklJYSOJVnRlFwQjq42HLw?e=MUC2Rn.
> > >
> > > As can be seen, NsDiff could help improving the uncertainty prediction by combining two mean/variance estimators. we would include the figure in the updated version to demonstrate the convergence.
> > >
> > >
> > > **Q11**: summarize Algorithm 1,2 into a combined Algorithm 3.
> > >
> > > **A11**: Thanks for this advice, we would provide a comprehensive Algorithm 3 in the updated version.
> > >
> > > **Q12**: paired-sample t-tests for ablation exp. in Table 5.
> > >
> > > **A12**: We thank the reviewer for this valuable suggestion. The results evident and well-supported by visualized results (Figures 3 and 4), which is why we didn’t initially consider statistical testing. To ensure reliability across experiments, we used identical seeds (1, 2, 3) throughout the paper. To address your concern, we support the statistical analysis with addtional seeds [1, 2, 3, 4, 5, 6] for the paired-sample t-tests. The results of these tests are summarized in the table below:
> > >
> > >
> > > | **Comparison**                        | **t-statistic** | **p-value**  |
> > > |------------------------------------|-------------|----------|
> > > | **CRPS: NsDiff vs w/o LSNM** | -3.4549 | 0.0181 |
> > > | **CRPS: NsDiff vs w/o UANS** | -3.9949 | 0.0104 |
> > > | **QICE: NsDiff vs w/o LSNM** | -3.0978 | 0.0269 |
> > > | **QICE: NsDiff vs w/o UANS** | -4.2117 | 0.0084 |
> > >
> > > Both CRPS and QICE comparisons for NsDiff against the ablation variants (w/o LSNM and w/o UANS) yielded statistically significant results, indicating a consistent performance advantage of the full NsDiff model. We will include these updated results in the revised version of the paper.

---

### Official Review · Reviewer_PZBU · 2025-03-17

**Overall Recommendation:** 4

**Summary:**

The paper introduces a novel diffusion-based probabilistic forecasting framework, called NsDiff, which is designed to address the non-stationary nature of uncertainty in time series data. Traditional Denoising Diffusion Probabilistic Models (DDPMs) typically rely on an Additive Noise Model (ANM) with fixed variance, limiting their ability to capture the dynamic uncertainty observed in many real-world applications. To overcome this limitation, the authors propose incorporating a Location-Scale Noise Model (LSNM) that allows the noise variance to vary with the input context. The authors also try to validate their results in a wide range of datasets with comparisons to some existing methods.

**Claims And Evidence:**

I think the claims made in the submission are clear and well-supported by the evidence detailed in the main text and the supplementary material. It is nice that the authors open-sourced their code.

**Essential References Not Discussed:**

I believe the paper would benefit from incorporating several additional essential references that provide valuable context and complementary perspectives. For example:

[1] Chen, Y., Goldstein, M., Hua, M., Albergo, M. S., Boffi, N. M., & Vanden-Eijnden, E. (2024). Probabilistic Forecasting with Stochastic Interpolants and Föllmer Processes. arXiv preprint arXiv:2403.13724.
This work is relevant because it also leverages diffusion models for probabilistic predictions, proposes innovative loss functions for optimizing noise schedules, and offers flexible alternatives for the base measure within diffusion models.

[2] Jiang, R., Lu, P. Y., Orlova, E., & Willett, R. (2023). Training Neural Operators to Preserve Invariant Measures of Chaotic Attractors. Advances in Neural Information Processing Systems, 36, 27645–27669.
This reference is pertinent as it addresses similar applications where noise induces non-stationarity in observations and presents methods to enhance long-term prediction accuracy.

Incorporating these references would strengthen the discussion by situating the proposed work within the broader context of recent advances in diffusion models and long-term forecasting under non-stationary conditions.

**Experimental Designs Or Analyses:**

I think the experiments done by the authors are impressive and comprehensive as they test the proposed method on many real-world datasets of different nature and dimensionalities.

**Methods And Evaluation Criteria:**

I think the proposed methods make sense for the applications of probabilistic forecasting of non-stationary systems.

**Other Comments Or Suggestions:**

N/A

**Other Strengths And Weaknesses:**

N/A

**Questions For Authors:**

I don't have any questions for authors.

**Relation To Broader Scientific Literature:**

I think the proposed method has a potential of being applied to many scientific applications, including weather forecasting, forecasting of physical systems (e.g. fluid dynamics), and epidemics.

**Theoretical Claims:**

I looked at the proof in Appendix A, which is straightforward and easy to follow.

---

> ### Author Rebuttal · Authors · 2025-03-31
>
> **Q1**: additional key references [1-2].
>
> **A1**: Thanks for your recognition of our work. We agree that the references [1–2] provide meaningful context and will help strengthen our discussion. In particular, we find several aspects of these works especially relevant to our setting:
>
>
> The work by Chen et al. [1] proposes a novel probabilistic forecasting framework based on Föllmer processes and stochastic interpolants. Their approach to learning noise schedules through tailored loss functions resonates with our motivation to adapt diffusion endpoints for better uncertainty modeling, especially under non-stationary conditions.
>
> Jiang et al. [2] address the challenge of long-term forecasting in chaotic systems by preserving invariant measures. Their use of contrastive learning to stabilize dynamics over time without requiring domain-specific priors is an inspiring direction that aligns with our interest in modeling non-stationary behavior robustly.
>
>
> We will cite and briefly discuss these works in the revised version to better contextualize our contributions within the broader landscape of diffusion-based and non-stationary forecasting techniques.
>
>
> [1] Chen, Y., Goldstein, M., Hua, M., Albergo, M. S., Boffi, N. M., & Vanden-Eijnden, E. (2024). Probabilistic Forecasting with Stochastic Interpolants and Föllmer Processes. arXiv preprint arXiv:2403.13724.
>
> [2] Jiang, R., Lu, P. Y., Orlova, E., & Willett, R. (2023). Training Neural Operators to Preserve Invariant Measures of Chaotic Attractors. Advances in Neural Information Processing Systems, 36, 27645–27669.

---

### Decision · Program_Chairs · 2025-05-01

**Decision:**

Accept (spotlight poster)

**Comment:**

The work under review proposes to explicitly model the time-dependent uncertainty inherent to time series forecasting in diffusion models. A location-scale noise model is used to flexibilize the usual simple additive noise model.

All reviewers agree about the soundness and value of this contribution. The method is theoretically sound, widely applicable, and extensively tested empirically. Given the relevance and prominence of diffusion-based methods, this work has potential to impact many others.

Please do include the point-based results (MSE, MAE) requested by multiple reviewers in the final version.